# On the Benefits of Public Representations for Private Transfer Learning under Distribution Shift

**Pratiksha Thaker**
Carnegie Mellon University
pthaker@andrew.cmu.edu

**Amrith Setlur**
Carnegie Mellon University
asetlur@andrew.cmu.edu

**Zhiwei Steven Wu**
Carnegie Mellon University
zstevenwu@andrew.cmu.edu

**Virginia Smith**
Carnegie Mellon University
smithv@andrew.cmu.edu

## Abstract

Public pretraining is a promising approach to improve differentially private model training. However, recent work has noted that many positive research results studying this paradigm only consider in-distribution tasks, and may not apply to settings where there is distribution shift between the pretraining and finetuning data—a scenario that is likely when finetuning private tasks due to the sensitive nature of the data. In this work, we show empirically across three tasks that even in settings with large distribution shift, where both zero-shot performance from public data and training from scratch with private data give unusably weak results, public features can in fact improve private training accuracy by up to 67% over private training from scratch. We provide a theoretical explanation for this phenomenon, showing that if the public and private data share a low-dimensional representation, public representations can improve the sample complexity of private training even if it is *impossible* to learn the private task from the public data alone. Altogether, our results provide evidence that public data can indeed make private training practical in realistic settings of extreme distribution shift.

## 1 Introduction

Learning models from user data can potentially disclose sensitive user information, violating privacy constraints [1–3]. Differential privacy is a standard framework that can be used when learning models from sensitive data to mitigate the risk of leaking private information [4]. However, differentially private learning may significantly degrade accuracy, which remains a barrier to adoption [5]. This has motivated recent works to explore the benefits of incorporating publicly available data into private training, e.g., by pretraining a model on public data and then finetuning it using private data. Empirically, this paradigm has been shown to substantially improve performance on private tasks relative to fully-private training [6–13].

While these results are encouraging, Tramèr et al. [14] point out that much of the existing work focuses on *in-distribution* tasks, where the public and private tasks are very similar. For example, many private vision models [15–19] use public features pretrained on ImageNet [20], CIFAR-10 or CIFAR-100 [21], but these works also simulate private *transfer* performance by finetuning on one of these datasets. In fact, Tramèr et al. [14] point out that "*every single* class contained in the CIFAR-10 dataset has an identical class label in the ImageNet dataset!" This is particularly problematic when attempting to understand the utility of public pretraining for private tasks, because in practice the private task is likely to contain sensitive data that is *not* perfectly represented by public data, such as in applications in medicine [22] or law [23]. Indeed, if data is already well-represented in a public dataset, the *zero-shot*

performance of a model trained only on public data should be good enough that no private "transfer" learning is required, potentially making these benchmark datasets uninformative for evaluating the benefits of transfer learning.

From a practical perspective, it is particularly important to understand transfer learning in the private setting: if a *non*-privacy-sensitive task is poorly represented by the pretrained features, one solution might be to simply add the data from that task into the public training dataset and learn a more general set of features for downstream use. But privacy-sensitive data cannot be used to train a public backbone, and individual private datasets often cannot be combined or shared. Thus, the ability to leverage public features to improve the sample dependence of private learning is critical.

**Our contributions.**    In this work, we provide evidence to alleviate these concerns, showing theoretically and empirically that public pretraining can be helpful even in settings with realistic and possibly extreme distribution shift between public (training) and private (transfer) tasks. In particular, we focus on concept shift, where the conditional distributions $P(Y \mid X)$ can vary drastically between public and private tasks. Our results are summarized as follows.

First, we conduct empirical case studies[1] on three datasets to show that public features improve private training accuracy even under extreme distribution shift. In particular, we use a pretrained CLIP ViT-B vision model for public features and measure the accuracy of private transfer learning on datasets including the PatchCamelyon (PCam) [24], Functional Map of the World (fMoW) [25], and Remote Sensing Image Scene Classification (RESISC45) [26]. On all three datasets, the pretrained model has unacceptably low zero-shot accuracy (random guessing on both PCam and fMoW), indicating that "perfect privacy" with zero-shot queries is likely hopeless. In comparison, on CIFAR-10, the CLIP ViT-B/32 model achieves 91.3% zero-shot accuracy [27], making transfer learning performance far less relevant as the zero-shot accuracy is already high. We observe that across all datasets, private finetuning and linear probing using public features outperform differentially training from scratch – by up to 67%. In addition, private linear probing consistently outperforms private finetuning.

Motivated by our empirical results, we provide a stylized theoretical model to understand and explain our findings. We study a simple linear transfer learning model, a common theoretical model in the non-private meta-learning literature [28–34], to show the statistical benefit of learning a shared, low-dimensional *representation* (in our model, a low-rank linear subspace) using public data. Our transfer learning model captures an extreme form of concept shift in the sense that the target model on private data is entirely different from those on public data, even though they are all contained in the same subspace. Analogous to the paradigm of public pre-training then private linear probing, we analyze a simple two-stage algorithm that (1) first estimates the shared, low-dimensional representation (or subspace) from a diverse set of tasks in public data, and (2) performs private linear regression within the learned subspace. By leveraging the dimensionality reduction, we provide a better sample complexity that scales with the rank of the shared subspace instead of the ambient dimension of the features. To complement this sample complexity bound, we also show a novel lower bound that shows that our bound is tight among algorithms that search for regression parameters within a fixed low-rank subspace estimate.

In short, our findings provide optimistic insights regarding the concerns raised by Tramèr et al. [14]. Specifically, Tramèr et al. [14] suggest that "current methods for large-scale pretraining may be less effective." In contrast, our results indicate that pretrained features can indeed benefit private learning, even under concept shift. Additionally, our findings address another concern from Tramèr et al. [14] regarding the necessity of uploading private data to cloud services for finetuning large models due to high resource requirements. We demonstrate that training a linear probe privately is more effective, potentially requiring significantly fewer resources (both memory and computation) than finetuning a full model.

## 2    Related Work

**Empirical studies of public pretraining for private learning.**    As Tramèr et al. [14] point out, existing empirical studies on public pretraining for private learning largely focus on transfer between similar datasets. For example, [15–19, 35] pretrain on CIFAR-100 or ImageNet and finetune on CIFAR-10 or STL-10 (a dataset very similar to CIFAR-10). [8] pretrains on Places365 and finetunes on ImageNet. [18, 36] pretrain on JFT and finetune on ImageNet. Finally, [37, 38, 9, 39] pretrain and finetune on publicly available text on the Web.

---

[1]Code will be made available at `https://github.com/pratiksha/private-transfer`.

All of these works build evidence that pretraining could be beneficial for private learning. Unfortunately, because the public and private tasks are so similar, these results are unlikely to be representative of real-world private training in which the private task requires learning a model on sensitive data with a very different distribution from data available on the Web.

Recent work [40] evaluates their algorithm on private learning tasks that are out-of-distribution for the feature extractor they use, including the PCam dataset that we also study. However, their algorithm requires access to (nearly) in-distribution *public* data in order to learn a projection matrix into a low-dimensional space. We argue that this is a strong and unrealistic assumption considering the arguments put forth in Tramèr et al. [14] that private data, because of its sensitive nature, will not be well-represented by public datasets. Our work instead focuses on understanding the improvements from using off-the-shelf feature extractors, with no in-distribution public data, over fully-private learning.

**Transfer or meta-learning.**  Our results build on the framework of Tripuraneni et al. [28] for nonprivate transfer learning with a low-dimensional subspace. This linear, low-dimensional subspace assumption has been studied extensively in the nonprivate meta-learning literature as a tractable model for real shared representation learning [28–34]. However, none of these works consider the setting of public subspace estimation followed by private transfer learning. PILLAR [40] makes a shared subspace assumption in the private setting, but on the input features rather than on the models.

**Private algorithms that leverage public data.**  A number of prior works have theoretically studied the benefits of public data in other settings, including mean estimation [41], query release [42–44], and optimization when *gradients* lie in a low-rank subspace [45–47]. Kairouz et al. [45] in particular gives a similar analysis using the principal angle error of the subspace, but the analysis does not apply directly as we assume that models, rather than gradients, lie in a shared low-dimensional subspace. As a result, the algorithm in that work requires expensive subspace oracle calls on every iteration and would be computationally suboptimal in our setting.

Finally, as discussed earlier, pretraining has empirically been shown to be useful in a number of domains, including vision [6–8] and NLP [9–12]. While our work does not model the complexities of neural networks, we can understand our results as a stylized version of finetuning in which the public network is tuned with linear regression on the last layer, potentially giving insight into these more complex models.

**Theoretical analyses of pretraining for private learning.**  Ganesh et al. [35] provides a lower bound construction for a related setting in which public data is abundant and the private task is out of distribution, though does not consider the case where the public and private task explicitly share structure. In our setting, learning from the public data alone provides no guarantees on the transfer task, as we do not assume any bounded shift in the data distributions or target parameters between the public tasks to the private tasks; the key information enabling more efficient learning is the shared structure among the tasks. PILLAR [40] incorporates public pretraining, but their analysis focuses on the benefits of dimensionality reduction using in-distribution public data, rather than transfer from out-of-distribution public data. Finally, Ke et al. [19] study the tradeoffs between linear probing and finetuning in the private setting. While their empirical results focus on the in-distribution image recognition settings outlined previously, their theoretical results corroborate our findings that even under extreme distribution shift, linear probing is more effective than finetuning under differential privacy.

## 3 Preliminaries

**Notation.**  Throughout the paper, we use lower-case $v$ for vectors, upper-case $V$ for matrices and calligraphic $\mathcal{V}$ for sets. Generally, we use the "hatted" notation $\hat{B}$, $\hat{\alpha}$ to refer to estimates of the underlying population variables. The use of $\mathcal{O}, \Omega, \Theta$ is standard and $\tilde{\mathcal{O}}, \tilde{\Omega}$ hides $\mathrm{polylog}$ factors in quantities we specify separately. We use $\|\cdot\|_F$ for Frobenius, $\|\cdot\|_{\mathrm{op}}$ for operator and $\|\cdot\|_p$ for $\ell_p$ norms.

### 3.1  Differential Privacy

Differential privacy (DP) is a quantitative constraint on the information gained from a released statistic [48]. Definition 3.1 restates the standard $(\varepsilon, \delta)$-differential privacy introduced in [4].

**Definition 3.1** ( $(\epsilon, \delta)$-differential privacy [4]).  *Given $\epsilon \geq 0$, $\delta \in [0,1]$ and a neighboring relation $\sim$, a randomized mechanism $M : \mathcal{X}^n \to \mathcal{Y}$ from the set of datasets of size $n$ to an output space $\mathcal{Y}$ is*

$(\epsilon, \delta)$-differentially private *if for all neighboring datasets $\mathcal{S} \sim \mathcal{S}' \subseteq \mathcal{X}$, and all events $E \subseteq \mathcal{Y}$,*

$$\Pr[\mathcal{M}(\mathcal{S}) \in E] \leq e^{\epsilon} \cdot \Pr[\mathcal{M}(\mathcal{S}') \in E] + \delta.$$

*Here, probabilities are taken over the random coins of $\mathcal{M}$.*

The "neighboring" relation differs according to the desired privacy guarantee. In this paper, we will study *row-level* privacy in which neighboring datasets $\mathcal{S} \sim \mathcal{S}'$ differ in a single element.

## 3.2    Problem Setting: Leveraging public samples for private transfer learning

We will study a setting in which the learner first sees $n_1$ public samples $(x_i, y_i)$, possibly drawn from multiple different underlying tasks (i.e., sample distributions) $P_1, \ldots, P_t$, and then sees $n_2$ private samples from a new task $P_{t+1}$. The goal is to learn a predictor $f : \mathbb{R}^d \to \mathcal{Y}$ that maps inputs $x \in \mathbb{R}^d$ to outputs $y \in \mathcal{Y}$ with the constraint that $f$ must satisfy $(\varepsilon, \delta)$-differential privacy. We aim to minimize the population loss on the private task:

$$\mathcal{L}(f) = \mathbb{E}_{(x,y) \sim P_{t+1}}[\ell(f(x), y)]. \tag{1}$$

The private learner may or may not use the public samples. We assume the samples are drawn i.i.d. conditioned on the task, but make no other assumptions on the task distribution or the number of samples drawn from each task. In Section 5, we develop a theoretical model of the relationship between the public and private tasks that allows the learner to effectively leverage information from the public tasks to improve private learning.

# 4    Public Data Improves Out-of-Distribution Private Transfer

We begin by studying three datasets and show empirically that public data can provide benefits for private transfer learning even when the public data alone gives unusable zero-shot results on the private task. Each of the tasks we evaluate on has unusably low zero-shot performance on CLIP [27], indicating that these are highly out-of-distribution relative to the pretraining data. This directly contrasts with existing work: the CLIP model that we use (pretrained with LAION-2B) achieves 66.6% zero-shot performance on ImageNet and 93.5% accuracy on CIFAR-10.

## 4.1    Datasets

**PatchCamelyon.**    The PatchCamelyon (PCam) medical images dataset is a binary lymph node tumor detection task highlighted by [14]. [14] point out that CLIP [27] as well as other similar text-vision models [22] have notably poor zero-shot performance on PCam: CLIP ViT-B/32 achieves 51.2%, or close to random, in our evaluation. The poor zero-shot performance (relative to tasks like ImageNet or CIFAR) indicates that the task is truly "out of distribution" in comparison to the source (public) data. Moreover, being medical image data, PCam more faithfully represents a highly privacy-sensitive dataset.

While the next two datasets are not medical image datasets, they are widely studied distribution shift datasets that have poor zero-shot performance on the training data, making them suitable for understanding transfer learning performance. In particular, they are remote sensing datasets; Wang et al. [49] analyze LAION-2B and find that only *0.03% of samples* are remote sensing images, another strong indication that this data is underrepresented in pretraining.

**fMoW.**    The Functional Map of the World (fMoW) dataset [25, 50] is a 62-class satellite image classification task. The pretrained CLIP ViT-B model achieves only 1.64% zero-shot accuracy, so "perfect privacy" with zero-shot classification is not possible.

**RESISC45.**    The Remote Sensing Image Scene Classification dataset [26] is a 45-class satellite image classification task. The pretrained CLIP ViT-B model achieves 56.3% zero-shot accuracy.

## 4.2    Experimental Setup

We train a ViT-B/32 model [51] on each dataset (which has output dimension 512) with a linear classification head for each task. For models trained from scratch, we use Xavier initialization on

|                          | PCam | fMoW | RESISC45 |
|--------------------------|------|------|----------|
| Zero-shot CLIP           | 51.2 | 1.64 | 56.3     |
| Full training from scratch | 78.2 | 19.7 | 41.9   |
| Full finetuning          | 82.5 | 58.2 | 93.6     |
| Linear probing           | 83.5 | 42.1 | 91.7     |

**Table 1:** Test accuracy of nonprivate training on each dataset that we evaluate.

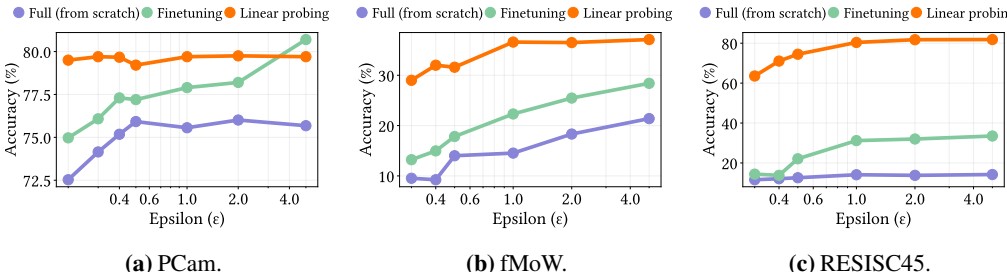

**(a)** PCam.  **(b)** fMoW.  **(c)** RESISC45.

**Figure 1:** Private training on three datasets. (a) PCam is a binary classification task on which private training from scratch achieves relatively high accuracy, but linear probing on the pretrained model still improves accuracy up to 4%. (b) The fMoW model trained from scratch is unusable at low privacy levels while linear probing achieves close to nonprivate accuracy. (c) On RESISC45, linear probing outperforms full finetuning by over 50% at all $\varepsilon$ levels.

the weights, while for pretrained features, we use OpenCLIP [52] models initialized with weights pretrained using LAION-2B (a 2B-sample subset of LAION-5B [53]). We use the Opacus library [54] to implement private training. For each training setting we performed a hyperparameter sweep over learning rate ($\{1e-6,...,1e-2\}$) and number of epochs (1-10 for full training and 1000-2500 for linear probing), and for private learning, clipping norm ($\{0.5,1.0,2.5,5.0\}$). For both private and nonprivate models, we evaluate training from scratch, full finetuning, and linear probing. We train private models for $\varepsilon \in \{0.3,0.4,0.5,1.0,2.0,5.0\}$ for each training setting. For PCam and RESISC45, we use SGD with momentum (parameter 0.9), while for fMoW we found that Adam gave better performance [55]. We use a cosine learning rate schedule for all experiments and a batch size of 32. Each finetuning run is performed on an A100 or A6000 GPU.

### 4.3   Results

We plot our private training results in Figure 1, and also provide nonprivate training and zero-shot CLIP numbers for reference in Table 1. Zero-shot CLIP has random accuracy on PCam (binary) and fMoW (62 classes). On RESISC45, zero-shot CLIP performs better than training from scratch (nonprivately), but finetuning and linear probing have nearly 40% higher accuracy. As pointed out by Tramèr et al. [14], if the zero-shot numbers (with no knowledge of the transfer task) matched the best performance of finetuning, then "perfect privacy" with no finetuning would be sufficient. But in each of these settings, the zero-shot performance is considerably worse than what is achievable with finetuning in both the nonprivate and private settings.

Across all datasets, we find that any type of finetuning significantly outperforms training privately from scratch. This indicates that the pretrained features are indeed contributing to training accuracy. Further, we find across all datasets

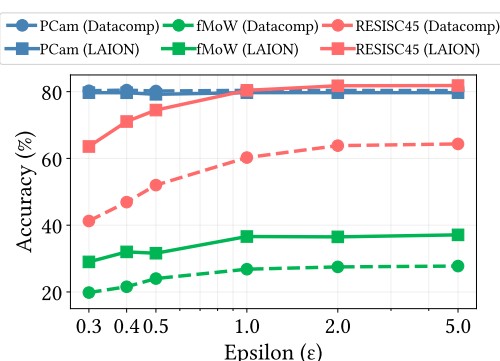

**Figure 2:** Linear probing results for ViT-B/32 pretrained on a 14M subset of Datacomp-1B and on LAION-2B. (Solid lines are LAION results while dashed lines are Datacomp results.) While the linear probing results in both settings outperform training from scratch, the worse accuracy on the Datacomp pretrained features are reflective of the lower-quality features from the smaller pretraining set.

that linear probing (fixing the pretrained features) outperforms full finetuning, sometimes by a large margin, as in the case of RESISC45. This finding is consistent with theoretical work [19] that models

the benefits of linear probing over finetuning under differential privacy. This is also consistent with earlier empirical findings on (in-distribution) private finetuning [8].

The key takeaway is positive: that features that work well for nonprivate transfer learning also benefit private transfer learning even when the distribution shift is large. While the conclusions are similar, these results are especially important in the private setting: training models from scratch with strong privacy is simply infeasible for many tasks, resulting in only around 10% test accuracy for fMoW and RESISC at small values of $\varepsilon$.

To further support our results, we additionally evaluate linear probing for all three datasets with features pretrained on a 14M subset of Datacomp-1B [56] in Figure 2. The trends in this setting are the same and linear probing still outperforms private training from scratch on all datasets, but the smaller pretraining dataset leads to lower-quality features that impact the final accuracy of linear probing.

## 5    Theoretical Model

Our empirical results show that even when distribution shift is extreme, public pretraining can indeed improve the accuracy of private training. In order to explain this observation, we study a simplified linear regression setting in which the goal is to estimate regression parameters privately for a single, unseen private task. This setting has been studied extensively in the nonprivate meta-learning literature as a theoretically tractable model to explain results on larger models [28–34], and we propose a novel extension to the private setting that helps explain our empirical findings.

We show that if the regression parameters for the private task lie in a low-dimensional subspace that is shared with the public tasks, the learner can use the public data to efficiently estimate the low-dimensional subspace, project the private data into the subspace, and thus achieve private estimation error rates that match optimal private linear regression rates (up to constant factors) in $k$ dimensions (rather than $d$ dimensions), with an additive term that accounts for the error in estimating the subspace publicly. These results hold even when we make no assumptions on the relationship between the public and private task other than that they share the same low-dimensional subspace.

We additionally provide a novel lower bound that shows that the algorithm we analyze for our upper bound achieves the optimal rate among "two-stage" algorithms that estimate the transfer parameters within a fixed low-dimensional subspace.

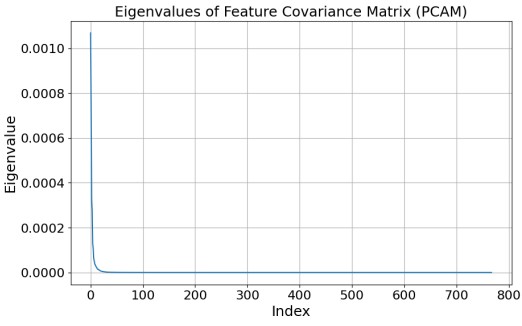

**Figure 3:** Eigenspectrum of feature covariance matrix for PCam features extracted from pretrained CLIP ViT-B/32 model.

**How realistic is the shared subspace assumption?**    As mentioned, the theoretical model we analyze has been previously studied to explain meta-learning results in nonprivate settings. Nevertheless, one might ask how realistic the model is for the particular settings we study, especially the assumption of a low-rank subspace shared by both the training and transfer tasks.

As a step toward understanding whether this assumption holds in practice, we plotted the eigenspectrum of the feature covariance matrix computed after extracting features of PCam images from the CLIP ViT-B-32 pretrained model (Figure 3).

From these results, we see that the pretrained features are approximately low-rank for the out-of-distribution task PCam, yet a linear probe over these features achieves good (83.5%) performance (Table 1). The fact that the representation still gives good performance when only a linear layer is trained on top suggests that the data does fundamentally lie in or near the low-rank space that is identified by the pretrained model.

In Appendix A, we plot and see similar results for the fMoW and RESISC45 datasets, where linear probing is similarly successful (relative to full finetuning).

## 5.1 Model and preliminaries

We first describe our model of the data distribution for the private task, learning objective, any assumptions we make and results from prior works we use.

### 5.1.1 Shared task structure

We consider linear regression models in which every observation $(x_i,y_i)$ for a given task is generated according to:

$$x_i \sim \mathcal{N}(0,I_d), \qquad \eta \sim \mathcal{N}(0,1)$$
$$y_i = x_i^\top B\alpha_{t(i)} + \eta_i. \tag{2}$$

The covariates $x_i$ and noise $\eta$ are sampled i.i.d. Here, $B \in \mathbb{R}^{d\times k}$ is an unknown, low rank $(k \ll d)$ feature matrix with orthonormal columns. The matrix $B$, and consequently the subspace spanned by its columns, is shared across all tasks in our problem setting. This includes both the public tasks that may be used to derive the initial estimate of $B$, as well as the private tasks in single-task and multi-task transfer settings.

The task vectors $\alpha_j$ are all assumed to lie in the true shared subspace $B$. $t(i)$ indexes the task $\alpha_j$ for the covariate $x_i$: public tasks are in $\alpha_{1...t}$, and the transfer task is $\alpha_{t+1}$. Note that the tasks are not random variables and we do not make distributional assumptions on the tasks for our results. In Appendix B we provide details on the requirements for the public tasks $\alpha_{1...t}$ (and also refer the reader to Tripuraneni et al. [28]), but for now we simply require that the public tasks are sufficiently "diverse" within $B$.

The learner sees $n_1$ samples from the public tasks (in total across all tasks) and $n_2$ samples drawn from the private task.

We are interested in learning $w$ that minimizes the following population risk:

$$\mathcal{L}(w) = \frac{1}{2}\mathbb{E}_{(x,y)}\left[(x^\top w - y)^2\right] \tag{3}$$

on the private task $B\alpha_{t+1}$.

### 5.1.2 Oracle for public subspace estimation

In stating our main results, we first assume access to an oracle that can output an orthonormal matrix $\hat{B} \in \mathbb{R}^{d\times k}$ that is "close to" $B$. We measure the distance between subspaces in terms of the principal angle distance, denoted $\sin\theta(B,\hat{B}) = \sin\theta(\hat{B},B)$ (see supplement and Tripuraneni et al. [28] for more discussion).

The following identities on $\sin\theta$ will be useful:

**Lemma 5.1** (subspace estimation errors). *The following inequalities are satisfied for matrices with orthonormal columns $B, \hat{B} \in \mathbb{R}^{d\times k}$ (and when $B, \hat{B}$ are swapped): $\|(I - \hat{B}\hat{B}^\top)B\|_F \geq \|(I - \hat{B}\hat{B}^\top)B\|_{op} = \sin\theta(\hat{B},B) \geq \|(I-\hat{B}\hat{B}^\top)B\|_F/\sqrt{k}.$*

**Instantiating the oracle with public data.** The following corollary characterizes the error incurred from estimating the underlying subspace from public data using the *method-of-moments* estimator from Tripuraneni et al. [28]. We state this bound for use in subsequent results but refer the reader to the supplement for the conditions required on public data in order to achieve this bound.

**Theorem 5.2** ([28], Theorem 3, simplified). *Let $A = (\alpha_1,...,\alpha_t)^\top$ be the public task matrix, $\nu = \sigma_k\left(\frac{A^\top A}{t}\right)$, and $\bar{\kappa} = \frac{tr(\frac{A^\top A}{t})}{k\nu}$ be the average condition number. If an equal number of samples is generated from each task, and $\bar{\kappa} \leq O(1)$ and $\nu \geq \Omega(\frac{1}{k})$, then the error of the method-of-moments estimator ( [28], Algorithm 1) is*

$$\sin\theta(\hat{B},B) \leq \tilde{O}\left(\sqrt{dk^2/n_1}\right). \tag{4}$$

*with probability at least $1 - O(n_1^{-100})$.*

We will refer to $\gamma \geq \sin\theta(B,\hat{B})$ as an upper bound on the error of the subspace estimation oracle. We give upper bounds with respect to $\gamma$ and also instantiate the bounds with the upper bound from Theorem 5.2.

**Algorithm 1** Two-phase algorithm for public-private linear regression using subspace estimation

---

**Input:** $n_1$ public samples drawn according to $(x_i, y_i)$, $x_i \sim \mathcal{N}(0, I_d)$, $y_i = x_i^\top B \alpha_{t(i)} + \eta$, $\eta_i \sim \mathcal{N}(0,1)$
      and $n_2$ private samples where $x_i, \eta_i$ have the same distribution, and $y_i = x_i^\top B \alpha_{t+1} + \eta_i$

  1: Use method-of-moments estimator ([28], Algorithm 1) to estimate $\hat{B}$ using public data
  2: Project private data $x_i$ to $k$-dimensional subspace: $x_i' = x_i^\top \hat{B}$
  3: Use DP-SGD variant of [57] on projected private data to estimate $\alpha_{t+1}$

**Output:** Parameter estimate $\hat{B}\hat{\alpha}_{t+1}$

---

### 5.1.3 Private linear regression in $d$ dimensions

We use in our analysis a known upper bound for private linear regression in $d$-dimensions. Theorem 5.3 states an informal result from [57] that upper bounds the excess risk for a variant of DP-SGD [15] (see Appendix B for more details). Furthermore, results from [58] imply that this upper bound is tight.

**Theorem 5.3** (Corollary 11 from [57], simplified). *Suppose we have $n_2$ i.i.d. datapoints $(x_i, y_i)$, where $x_i \sim \mathcal{N}(0, I_d)$ and $y_i = x_i^\top w + \epsilon_i$, and $\epsilon_i \sim (0, \sigma^2)$. Given sufficient private samples $n_2$, there exists an $(\varepsilon, \delta)$ private estimate $\hat{w}_{\mathrm{priv}}$ such that, with high probability:*

$$\mathcal{L}(\hat{w}_{\mathrm{priv}}) - \mathcal{L}(w) \lesssim \frac{d\sigma^2}{n_2}\left(1 + \tilde{\mathcal{O}}\left(\frac{d\log(1/\delta)}{n_2 \varepsilon^2}\right)\right). \tag{5}$$

### 5.2 Private transfer learning for a single task

**Algorithm.** Our proposed algorithm (Algorithm 1) first projects $x$ into the estimated subspace $\hat{B}_{\mathrm{pub}}$, i.e., $x \mapsto \hat{B}_{\mathrm{pub}}^\top x$, and then runs private linear regression in the $k$-dimensional subspace. This is analogous to linear probing in our experiments, which first uses the public encoder to compute a low-dimensional feature representation of the data and then learns a linear model using the features. While full finetuning of the model is also a common paradigm in the transfer learning literature, we point to [19] which shows that when the feature representation is sufficiently informative, linear probing outperforms finetuning under differential privacy – a result that supports our empirical findings.

The following theorem states that Algorithm 1 achieves a rate that matches optimal rates for private linear regression in $k$-dimensions, up to the subspace estimation error $\gamma$.

**Theorem 5.4** (single-task private transfer upper bound). *Assume we have access to a subspace estimation oracle that solely uses public samples to provide estimate $\hat{B}_{\mathrm{pub}}$ for the unknown subspace $B$ of a private task defined by the pair $(B, \alpha_{t+1})$ in (2). Further, the estimate satisfies $\sin\theta(\hat{B}_{\mathrm{pub}}, B) \leq \gamma$. Given $n_2$ i.i.d. samples from the distribution of this private task, Algorithm 1 outputs an estimate $\hat{B}_{\mathrm{pub}}\hat{\alpha}_{t+1}$ that is $(\varepsilon, \delta)$-differentially private, and with high probability incurs a risk of:*

$$\mathcal{L}(\hat{B}_{\mathrm{pub}}\hat{\alpha}_{t+1}) - \mathcal{L}(B\alpha_{t+1}) \tag{6}$$

$$\leq \tilde{O}\left(\|\alpha_{t+1}\|_2^2(\gamma^2 + 1)\right)\tilde{O}\left(\frac{1}{n_2^{100}} + \frac{k}{n_2} + \frac{k^2\log(1/\delta)}{n_2^2\varepsilon^2}\right) + \gamma^2. \tag{7}$$

*Proof sketch.* The proof nearly follows from existing bounds on subspace estimation and private linear regression. The key difficulty is that regression on the input $x \sim \mathcal{N}(0, I_d)$ projected into the estimated subspace $\hat{B}_{\mathrm{pub}}$ still leaves the residual that does not lie in $\hat{B}_{\mathrm{pub}}$, which can be treated as a noise term if we can show that the residual is independent of the projected $x$. We can show this because $\hat{B}_{\mathrm{pub}}$ is orthogonal to $\hat{B}_{\mathrm{pub}}^\perp$ (spans null space of $\hat{B}_{\mathrm{pub}}$), so under the i.i.d. Gaussian assumption on $x$, the residual is independent of the projected $x$. As a result, we obtain the private linear regression rate in $k$ dimensions with a variance of $1 + \gamma^2$ rather than 1 and an additive $\gamma^2$ bias.

**Discussion.** From Theorem 5.4, we can break down the errors into an unavoidable bias due to the subspace estimation error (dependent only on the number of public samples) and the subsequent linear regression error due to privacy. For a subspace estimation error $\gamma$ we require $n_1 \geq \frac{dk^2}{\gamma^2}$. Given this inevitable error we can hope to achieve an accuracy of $\mathrm{err} + \gamma^2$ where $\mathrm{err}$ is the additional linear

regression error and $\sin\theta(B,\hat{B}_{\mathrm{pub}})\leq\gamma$. This requires approximately:

$$n_2 \geq \frac{k}{\mathrm{err}} + \frac{k}{\varepsilon\sqrt{\mathrm{err}}} \tag{8}$$

samples. That is, if the subspace estimation error is zero then we achieve the rate of private linear regression in $k$ dimensions, and consequently optimal non-private rates when $\epsilon\to\infty$.

## 5.3 Lower bound for two-phase estimator

In the previous subsection, we proved an upper bound on the single-task transfer for row-level $(\varepsilon,\delta)$-DP private algorithm, when the publicly estimated subspace $\hat{B}_{\mathrm{pub}}$ is $\gamma$ accurate. In this section, we show that our upper bound is tight among algorithms for our problem that search for solutions within a fixed subspace.

In particular, we analyze the lowest possible transfer error achieved by any $(\varepsilon,\delta)$-DP algorithm that: (i) takes as input private dataset $\mathcal{S}$ of $n_2$ i.i.d. samples from task $\alpha_{t+1}$, $\gamma$-accurate public estimate $\hat{B}_{\mathrm{pub}}$, and (ii) outputs an estimate in the column space of $\hat{B}_{\mathrm{pub}}$. In Theorem 5.5, we present a lower bound on the risk suffered by any algorithm in such a class.

**Theorem 5.5** (Two-stage single-task private transfer lower bound)**.** *Let $M$ be an $(\varepsilon,\delta)$-DP private algorithm where $\varepsilon\in(0,1)$, $\delta<1/n^{1+\omega}$, $\omega>0$, that takes as input: (i) publicly estimated subspace $\hat{B}_{\mathrm{pub}}$ from an oracle that only uses public samples; and (ii) a dataset $\mathcal{S}$ of $n_2$ private samples. For any such $M$, there exists a private problem instance given by the pair $(B,\alpha_{t+1})$ where $B\in\mathrm{Gr}_{k,d}(\mathbb{R}),\alpha_{t+1}\in\mathbb{R}^k$, $\sin\theta(B,\hat{B}_{\mathrm{pub}})\leq\gamma$, and $\|B\alpha_{t+1}\|_2\leq1$, such that for $S$ sampled i.i.d. from this instance using the model in* (2)*, we have:*

$$\mathbb{E}_M\mathbb{E}_{\mathcal{S}|B,\alpha_{t+1}}\mathbb{E}_{(x,y)|B,\alpha_{t+1}}(y-M(\mathcal{S},\hat{B}_{\mathrm{pub}})^\top x)^2 \tag{9}$$

$$= \Omega\left(\left(\frac{k^2}{n_2^2\varepsilon^2}+\frac{k}{n_2}\right)(\sigma^2+\gamma^2)+\gamma^2\right). \tag{10}$$

*Proof Sketch.* Our proof relies mainly on tracing attacks in [59, 58], but our analysis additionally needs to handle the misspecification of the subspace $B$ which influences the construction of the worst case problem instance. When we project inputs $x\mapsto\hat{B}_{\mathrm{pub}}^\top x$, we can show that the projected samples can now be treated as i.i.d. samples from a $k$-dimensional linear regression model with independent noise. For a fixed $\hat{B}_{\mathrm{pub}}$, any choice of $B,\alpha_{t+1}$ affects both the scaling of the noise ($\propto\|(I-\hat{B}_{\mathrm{pub}}\hat{B}_{\mathrm{pub}}^\top)B\alpha_{t+1}\|_2^2$), and the direction of the regression vector, based on how much of the true parameter $B\alpha_{t+1}$ is captured in given subspace $\hat{B}_{\mathrm{pub}}$. To handle this, we first construct subclasses of the adversary, where each subclass fixes the norm of $\|\hat{B}_{\mathrm{pub}}^\top B\alpha_{t+1}\|_2$. Then, we lower bound the minimax risk over this subclass by via a Bayes risk which we further lower bound by constructing a *tracing adversary*.

We show that there exists a prior $\pi$ over $B\alpha_{t+1}$ where the probability of the intersection of the following two events is very low: (i) small estimation error $\mathbb{E}_\pi\mathcal{L}(M(\mathcal{S},\hat{B}_{\mathrm{pub}}))$, and (ii) small success rate for the tracing adversary to infer the membership of some element in $\mathcal{S}$. Since, $M$ has to be $(\epsilon,\delta)$ private, this reults in a Bayes risk lower bound.

**Discussion.** Our lower bound for the class of two-stage algorithms matches our upper bound in Theorem 5.4. This implies that our Algorithm 1 is optimal when $\hat{B}_{\mathrm{pub}}$ is the estimate given by the optimal subspace estimation oracle over public samples. When we use Algorithm 1 from [28], the estimation error matches lower bounds (Theorem 5 in [28]) upto a factor of $\sqrt{k}$.

## 5.4 Simulated results

Finally, we complement the results in this section through a simulated empirical study matching the setup described in Section 5.1.

**Setup.** We simulate $n_1$ samples $(x_i,y_i)$ from $t=100$ public tasks where the true dimension $d=25$ but the underlying subspace $B$ has rank 5. As baselines, we compare against nonprivate linear regression, DP-SGD without a subspace estimate, and DP-SGD initialized with the true subspace $B$, and compare against DP-SGD initialized with the subspace estimated using the method-of-moments estimator [28]. We use the Google Tensorflow implementation of DP-SGD for private learning [60].

We used a grid search of hyperparameters to set the clipping norm to $0.5$, learning rate to $0.1$, and used $50$ epochs of training for DP-SGD. We use the RDP accountant to set $\varepsilon = 1.1$ and $\delta = 1e-5$.

Our results are shown in Figure 4. We observe that, as expected, private training from scratch has high error, and additional public data ($n_1 = 500$ vs $n_1 = 2000$) improves performance, reducing the $\ell_2$ parameter error close to that of using DP-SGD with the true underlying subspace B (matching our intuition, for example, from Figure 2). However, we also see that when performing private transfer there are diminishing returns for this more precise subspace estimation, as the noise introduced via private learning becomes a dominating factor.

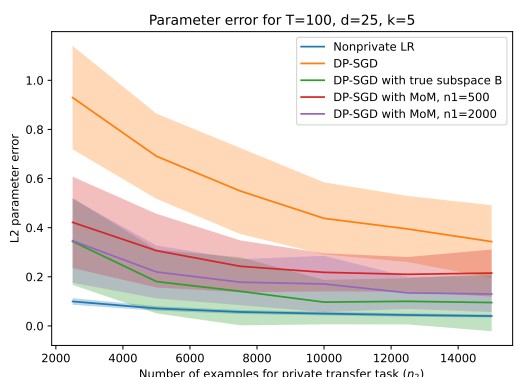

**Figure 4:** Empirical verification of setup described in Section 5.1.

## 6   Discussion and Limitations

Our results answer questions posed by [14] positively. Empirically, we show that across three datasets with significant shift between the public and private tasks, publicly pretrained features *do* make private learning far more effective, taking models from unusable when trained from scratch to close-to-nonprivate performance when trained privately with linear probing. In addition, we provide a theoretical model to explain our findings, based on models of nonprivate transfer learning. Our model supports our empirical findings, suggesting that public features should indeed reduce private sample complexity under even extreme distribution shift when the public and private tasks share a low-dimensional representation. Altogether, our conclusions are optimistic and provide confidence that public data can indeed support private training even for highly sensitive tasks that cannot and should not be used in public training. However, our linear subspace model has the clear limitation of being a simplified model for the neural network representations used in practice. As this is a limitation shared by literature on nonprivate transfer learning [28–34], improvements in this area would contribute to both the private and nonprivate transfer learning literature.

**Acknowledgements.**   Thanks to Shengyuan Hu, Tian Li, Qi Pang, and Anirudh Sivaraman for helpful discussions and feedback that improved the writing. This work was supported in part by the National Science Foundation grants IIS2145670 and CCF2107024, and funding from Amazon, Apple, Google, Intel, Meta, and the CyLab Security and Privacy Institute. Any opinions, findings and conclusions or recommendations expressed in this material are those of the author(s) and do not necessarily reflect the views of any of these funding agencies. Z.S.W. was in part supported by NSF Awards #1763786 and #2339775.

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

# A    Empirical evidence for shared subspace assumption

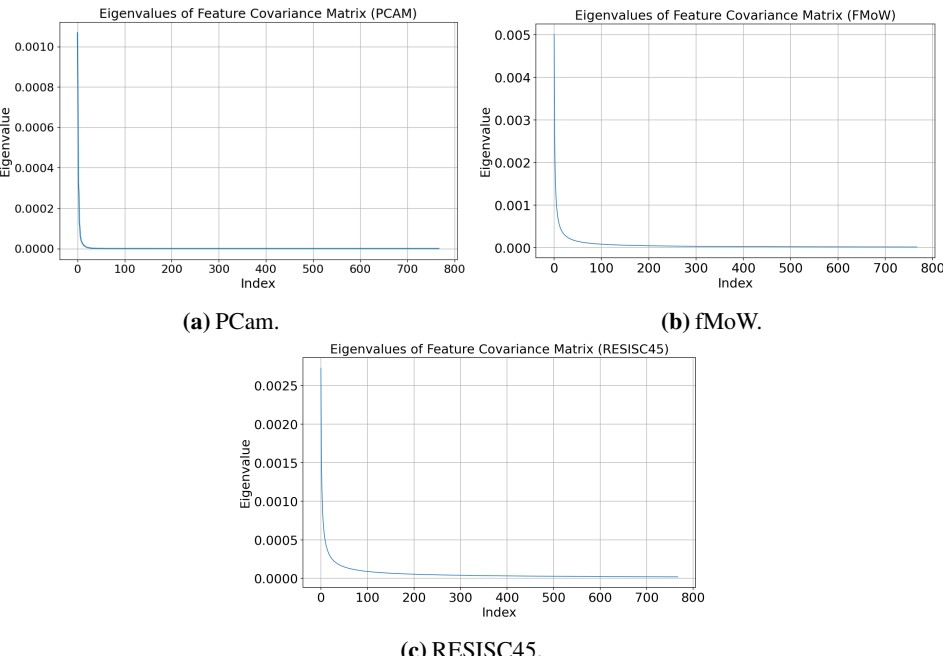

**(a)** PCam.
**(b)** fMoW.

**(c)** RESISC45.

**Figure 5:** Eigenspectra of feature covariance matrices for features extracted from pretrained CLIP ViT-B/32 model.

It is natural to ask whether the assumption that public (pretraining) tasks and private tasks truly share a low-dimensional subspace as we model in our theoretical analysis. In order to validate this, in figure 5, we plot the eigenspectra of the feature covariance matrices for each of the datasets we evaluate in Section 4. The key takeaway is that even though the three datasets are out of distribution for the pretraining data, the extracted features are low rank. In addition, these features are effective when used to train a linear probe (in contrast, if the subspace were misspecified, the features may be low-rank but lead to poor results with linear probing).

# B    Additional definitions and assumptions

## B.1    Preliminaries for Tripuraneni et al. [28]

In this section, we elaborate on the assumptions required to use algorithms from Tripuraneni et al. [28] for subspace estimation using *public* data (i.e., instantiating the subspace oracle).

### B.1.1    Principal angles

Our analysis requires a notion of distance between subspaces, for which we use the maximum principal angle [61]. We give a definition here and refer the reader to [62] or [63], Appendix A, for more details.

**Definition B.1** (Maximum principal angle)**.** *Let $U, V \in \mathbb{R}^{d \times k}$ be orthogonal matrices. Let $\mathcal{U}$ and $\mathcal{V}$ be the subspaces spanned by the columns of $U$ and $V$ respectively. The* maximum principal angle *$\theta \in [0, \pi/2]$ between $\mathcal{U}$ and $\mathcal{V}$ is defined by $\sin\theta(U, V) = \|UU^\top - VV^\top\| = \|U^\top V_\perp\| = \|V^\top U_\perp\|$.*

### B.1.2    Task diversity assumptions

In our model each data point $(x_i, y_i)$ is associated with a task $\alpha_{t(i)} \in \mathbb{R}^k$. We do not make distributional assumptions on these tasks, but estimating the subspace accurately requires certain diversity assumptions on the tasks. We inherit the following assumption from [28]:

**Assumption B.2** (Task diversity and normalization). *Define* $A = (\alpha_1, ..., \alpha_t)^\top$ *and* $\nu = \sigma_r\left(\frac{A^\top A}{t}\right)$. *The* $t$ *underlying task parameters* $\alpha_j$ *satisfy* $\|\alpha_j\| = \Theta(1)$ *for all* $j \in [t]$. *Moreover, we assume* $\nu > 0$.

In the following, we will also use the average condition number $\bar{\kappa} = \frac{\text{tr}\left(\frac{A^\top A}{t}\right)}{r\nu}$, and the worst-case condition number $\kappa = \sigma_1\left(\frac{A^\top A}{t}\right)/\nu$, to further characterize the task diversity.

Then we have:

**Theorem 5.2** ([28], Theorem 3, simplified). *Let* $A = (\alpha_1, ..., \alpha_t)^\top$ *be the public task matrix,* $\nu = \sigma_k\left(\frac{A^\top A}{t}\right)$, *and* $\bar{\kappa} = \frac{\text{tr}\left(\frac{A^\top A}{t}\right)}{k\nu}$ *be the average condition number. If an equal number of samples is generated from each task, and* $\bar{\kappa} \leq O(1)$ *and* $\nu \geq \Omega(\frac{1}{k})$, *then the error of the method-of-moments estimator (* [28], Algorithm 1) is*

$$\sin\theta(\hat{B}, B) \leq \tilde{O}\left(\sqrt{dk^2/n_1}\right). \tag{4}$$

*with probability at least* $1 - O(n_1^{-100})$.

## B.2 Estimation error bounds

We restate the full bound given by [57] for private SGD.

**Theorem B.3** ([57], Corollary 11, simplified). *Suppose we have data* $(x_i, y_i)$ *such that* $x_i \sim \mathcal{N}(0, I_k)$ *and* $y_i = x_i^\top w^* + \epsilon_i$, *where* $\epsilon_i \sim (0, \sigma^2)$. *Then, assuming* $n_2 \geq \tilde{\Omega}\left(k(1 + \frac{\sqrt{\log(1/\delta)}}{\varepsilon})\right)$, *we have:*

1. *Algorithm DP-AMBSSGD ([57], Algorithm 2) with parameters* $\eta = 1/4k$, $\alpha = \frac{\sqrt{8\log(1/\delta)}}{\varepsilon}$ *is* $(\varepsilon, \delta)$-*DP.*

2. *The output* $\hat{w}^{priv}$ *satisfies the following risk bound:*

$$\mathcal{L}(\hat{w}^{priv}) - \mathcal{L}(w^*) \leq \frac{\|w^*\|_2^2}{n_2^{100}} + \frac{8k\sigma^2}{n_2}\left(1 + \tilde{O}\left(\frac{k\log(1/\delta)}{n_2\varepsilon^2}\right)\right) \tag{11}$$

*with probability* $1 - O(n_2^{-100})$.

## C Technical lemmas

In this section we state or restate key lemmas that we will refer to in the following proofs.

We will use the following lemma to argue that we can project $x$ into the estimated subspace $\hat{B}$ and treat the residual (that lies outside of $\hat{B}$) as i.i.d. Gaussian noise.

**Lemma C.1** (Independence of $x$ residual). *Consider orthonormal matrices* $B$ *and* $\hat{B} \in \mathbb{R}^{d \times k}$ *and* $\alpha \in \mathbb{R}^k$. *Let* $(x, y)$ *be generated according to the model in Equation 2 where* $x \sim \mathcal{N}(0, I_d)$ *and* $y = x^\top B\alpha + \eta$. *Then the projection of* $x$ *into* $\hat{B}$, $x^\top(\hat{B}\hat{B}^\top)B\alpha$, *is independent of the residual that lies in the complement of* $\hat{B}$, *i.e.* $x^\top(I_d - \hat{B}\hat{B}^\top)B\alpha$. *Moreover, this residual is i.i.d. Gaussian.*

*Proof.* We can rewrite the distribution of $y \mid x$ in terms of the projection of the regression vector $B\alpha$ on to the column span of $\hat{B}$, when the input $x$ is also projected in the following way: $x \mapsto \hat{B}x$:

$$\begin{aligned} y &= x^\top B\alpha + \eta; \\ &= x^\top((\hat{B}\hat{B}^\top)B\alpha + (I_d - \hat{B}\hat{B}^\top)B\alpha) + \eta; \\ &= x^\top(\hat{B}\hat{B}^\top B\alpha) + x^\top(I_d - \hat{B}\hat{B}^\top)B\alpha + \eta; \\ &= (x^\top\hat{B})\hat{\alpha} + x^\top(I_d - \hat{B}\hat{B}^\top)B\alpha + \eta, \end{aligned} \tag{12}$$

where $\hat{\alpha} := \hat{B}^\top B\alpha$ is a $k-$dimensional vector in the column span of the given subspace $\hat{B}$.

Next, we note that the projection of input $x$ in the column span of $\hat{B}$ and its projection into the corresponding null space are independent, i.e., $x^\top(\hat{B}\hat{B}^\top) \perp\!\!\!\perp x^\top(I_d - \hat{B}\hat{B}^\top)$. This is easy to show since both $x^\top(\hat{B}\hat{B}^\top)$ and $x^\top(I_d - \hat{B}\hat{B}^\top)$ are jointly Gaussian and are marginally distributed as zero mean Gaussian random variables with their covariance:

$$\mathbb{E}[(\hat{B}\hat{B}^\top)xx^\top(I_d - \hat{B}\hat{B}^\top)]$$
$$= (\hat{B}\hat{B}^\top)\mathbb{E}[xx^\top](I_d - \hat{B}\hat{B}^\top)$$
$$= (\hat{B}\hat{B}^\top)I_d(I_d - \hat{B}\hat{B}^\top)$$
$$= \hat{B}\hat{B}^\top - \hat{B}\hat{B}^\top = 0.$$

This implies independence of the two projections. Note that the last step in the above calculation uses the fact that $\hat{B}^\top\hat{B} = I_k$. Since the two projections are independent, we can rewrite the conditional distribution $y \mid x$ as:

$$x^{\hat{B}} =: x^\top\hat{B}$$
$$y \mid x^{\hat{B}} \sim \mathcal{N}((x^{\hat{B}})^\top\hat{\alpha}, \hat{\sigma}^2), \quad \text{where, } \hat{\sigma}^2 = \sigma^2 + \|(I_d - \hat{B}\hat{B}^\top)B\alpha\|_2^2.$$

$\square$

**Lemma C.2** ([58, 64, 65]). *Let $M$ be an $(\varepsilon, \delta)$-differentially private algorithm with $0 < \varepsilon < 1$ and $\delta > 0$. Further, let $A_i = A_{\hat{\alpha}}((y_i, x_i^{\hat{B}}), M(\mathcal{S}))$ and $A_i' = A_{\hat{\alpha}}((y_i, x_i^{\hat{B}}), M(\mathcal{S}_i'))$ when $(y_i, x_i^{\hat{B}}) \in \mathcal{S}$ and $\mathcal{S}_i'$ replaces $(y_i, x_i^{\hat{B}})$ with another IID draw from the same distribution. Then, for every $T > 0$,*

$$\mathbb{E}A_i \leq \mathbf{E}A_i' + 2\varepsilon\mathbb{E}|A_i'| + 2\delta T + \int_T^\infty \mathbb{P}(|A_i| > t). \tag{13}$$

*Proof.* let $Z^+ = \max(Z, 0)$ and $Z^- = -\min(Z, 0)$ denote the positive and negative parts of random variable $Z$ respectively. We have

$$\mathbb{E}A_i = \mathbb{E}A_i^+ - \mathbb{E}A_i^- = \int_0^\infty \mathbb{P}(A_i^+ > t)\,dt - \int_0^\infty \mathbb{P}(A_i^- > t)\,dt.$$

For the positive part, if $0 < T < \infty$ and $0 < \varepsilon < 1$, we have

$$\int_0^\infty \mathbb{P}(A_i^+ > t)\,dt = \int_0^T \mathbb{P}(A_i^+ > t)\,dt + \int_T^\infty \mathbb{P}(A_i^+ > t)\,dt$$
$$\leq \int dt_0^T \left(e^\varepsilon\mathbb{P}(A_i^+ > t) + \delta\right)dt + \int_T^\infty \mathbb{P}(A_i^+ > t)\,dt$$
$$\leq \int_0^\infty \mathbb{P}(A_i'^+ > t)\,dt + 2\varepsilon\int_0^\infty \mathbb{P}(A_i'^+ > t)\,dt + \delta T + \int_T^\infty \mathbb{P}(|A_i| > t)\,dt.$$

Similarly for the negative part,

$$\int_0^\infty \mathbb{P}(A_i^- > t)\,dt = \int_0^T \mathbb{P}(A_i^- > t)\,dt + \int_T^\infty \mathbb{P}(A_i^- > t)\,dt$$
$$\geq \int_0^T \left(e^{-\varepsilon}\mathbb{P}(A_i'^- > t) - \delta\right)dt + \int_T^\infty \mathbb{P}(A_i^- > t)\,dt$$
$$\geq \int_0^T \mathbb{P}(A_i'^- > t)\,dt - 2\varepsilon\int_0^T \mathbb{P}(A_i'^- > t) - \delta T + \int_T^\infty \mathbb{P}(A_i^- > t)\,dt$$
$$\geq \int_0^\infty \mathbb{P}(A_i'^- > t)\,dt - 2\varepsilon\int_0^\infty \mathbb{P}(A_i'^- > t) - \delta T.$$

It then follows that

$$\mathbb{E}A_i \leq \int_0^\infty \mathbb{P}(A_i'^+ > t)\,dt - \int_0^\infty \mathbb{P}(A_i'^- > t)\,dt + 2\varepsilon\int_0^\infty \mathbb{P}(|A_i'| > t)\,dt + 2\delta T + \int_T^\infty \mathbb{P}(|A_i| > t)\,dt$$
$$= \mathbb{E}A_i' + 2\varepsilon\mathbb{E}|A_i| + 2\delta T + \int_T^\infty \mathbb{P}(|A_i| > t)\,dt.$$

$\square$

**Lemma C.3** (Stein's Lemma). *Let $Z$ be distributed according to some density $p(z)$ that is continuously differentiable with respect to $z$ and let $h: \mathbb{R} \to \mathbb{R}$ be a differentiable function such that $\mathbf{E}|h'(Z)| < \infty$, then:*

$$\mathbf{E}h'(Z) = \mathbf{E}\left[\frac{-h(Z)p'(Z)}{p(Z)}\right].$$

# D  Proofs for Section 5

## D.1  Proof of Theorem 5.4

In this section we prove the upper bound result on the two-phase algorithm for single-task transfer learning, which first estimates the subspace publicly, projects the inputs into the estimated subspace and then privately performs linear regression on the projected data.

*Proof.* Let $\sin\theta(\hat{B}, B) \le h(n_1)$ and $\mathbf{E}[(\langle\hat{\alpha}^{priv}, \hat{B}^\top x\rangle - y)^2] - \min_\alpha \mathbf{E}[(\langle\alpha, \hat{B}^\top x\rangle - y)^2] \le g(n_2)$.

Let $\hat{\alpha} = \min_\alpha \mathbf{E}[(\langle\alpha, \hat{B}^\top x\rangle - y)^2]$ (the best $\alpha$ using $x$ projected into $\hat{B}$), and let $\hat{\alpha}^{priv}$ be the output of DP-AMBSSGD on $x$ projected into the estimated $\hat{B}$. then we have

$$\mathbf{E}\left[\left(\langle\hat{\alpha}^{priv}, \hat{B}^\top x\rangle - y\right)^2\right] - \mathbf{E}\left[\left(\langle\alpha^*, B^\top x\rangle - y\right)^2\right]$$

$$= \mathbf{E}\left[\left(\langle\hat{\alpha}^{priv}, \hat{B}^\top x\rangle - y\right)^2\right] - \mathbf{E}\left[\left(\langle\alpha^*, B^\top x\rangle - y\right)^2\right] + \mathbf{E}\left[\left(\langle\hat{\alpha}, \hat{B}^\top x\rangle - y\right)^2\right] - \mathbf{E}\left[\left(\langle\hat{\alpha}, \hat{B}^\top x\rangle - y\right)^2\right].$$

We can break this into two parts: we will first bound

$$\mathbf{E}\left[\left(\langle\hat{\alpha}^{priv}, \hat{B}^\top x\rangle - y\right)^2\right] - \min_\alpha \mathbf{E}\left[\left(\langle\alpha, \hat{B}^\top x\rangle - y\right)^2\right] \tag{14}$$

and then

$$\mathbf{E}\left[\left(\langle\hat{\alpha}, \hat{B}^\top x\rangle - y\right)^2\right] - \mathbf{E}\left[\left(\langle\alpha^*, B^\top x\rangle - y\right)^2\right]. \tag{15}$$

We first bound (14). Note that according to the model (2),

$$y = x^\top B\alpha^* + \epsilon \tag{16}$$

where $\epsilon$ is $\mathcal{N}(0, 1)$.

However, our algorithm first projects $x$ into the space of the estimated $\hat{B}$ before performing linear regression in the lower-dimensional space, which introduces additional error.

We can rewrite $y$ as:

$$y = x^\top \hat{B}\hat{B}^\top B\alpha^* + x^\top (I - \hat{B}\hat{B}^\top)B\alpha^* + \epsilon \tag{17}$$

decomposing the first term into the projection into $\hat{B}$ and the remaining error due to projection.

By Lemma C.1, this residual term is independent of the first term (with $x$ projected into $\hat{B}$) and $\epsilon$.

We claim that the variance of the residual is $\sin(\theta)^2 + \|\alpha^*\|_2^2$:

Let

$$\epsilon' = x^\top (I - \hat{B}\hat{B}^\top)B\alpha^* + \epsilon \tag{18}$$

$$= x^\top \hat{B}_\perp \hat{B}_\perp^\top B\alpha^* + \epsilon \tag{19}$$

and note that the total variance is the sum of the variances because the terms are independent. Moreover, the first term is a rescaled i.i.d. Gaussian with zero mean. Then the variance of $\epsilon'$ is

$$\mathbf{E}[(x^\top \hat{B}_\perp \hat{B}_\perp^\top B\alpha^*)^\top x^\top \hat{B}_\perp \hat{B}_\perp^\top B\alpha^*]+\sigma^2$$
$$=\mathbf{E}[\alpha^{*\top} B^\top \hat{B}_\perp \hat{B}_\perp^\top xx^\top \hat{B}_\perp \hat{B}_\perp^\top B\alpha^*]+\sigma^2$$
$$=\mathbf{E}[\alpha^{*\top} B^\top \hat{B}_\perp \hat{B}_\perp^\top B\alpha^*]+\sigma^2$$
$$=\sin(\theta)^2\|\alpha^*\|_2^2+\sigma^2.$$

Moreover, we assume $\sigma^2=1$ so we have $\mathrm{var}(\epsilon')=\sin(\theta)^2\|\alpha^*\|_2^2+1$.

Using the rewritten $y$, we can treat the new private regression problem as estimating $\hat{B}^\top B\alpha^*$. Thus we will instantiate $g(n_2)$ with the linear regression bound from Theorem B.3 with $k$ dimensions and variance $\sigma^2=\sin(\theta)^2\|\alpha^*\|_2^2+1$.

Now we bound the second half of the expression,

$$\mathbf{E}\left[\left(\langle\hat{\alpha},\hat{B}^\top x\rangle-y\right)^2\right]-\mathbf{E}\left[\left(\langle\alpha^*,B^\top x\rangle-y\right)^2\right]$$
$$=\mathbf{E}\left[\left(\langle\hat{\alpha},\hat{B}^\top x\rangle-\langle\alpha^*,B^\top x\rangle+\langle\alpha^*,B^\top x\rangle-y\right)^2\right]-\mathbf{E}\left[\left(\langle\alpha^*,B^\top x\rangle-y\right)^2\right]$$
$$=\mathbf{E}\left[(\langle\hat{B}\hat{\alpha}-B\alpha^*,x\rangle)^2\right]=\|\hat{B}\hat{\alpha}-B\alpha^*\|_2^2$$

Finally this leaves us to bound $\|\hat{B}\hat{\alpha}-B\alpha^*\|_2^2$. We will make use of the following lemma:

**Lemma D.1.** *Let $\hat{\alpha}$ be the (public) linear regression estimate of the task $\alpha$ on the projected data $\hat{B}^\top x$. Then $\|\hat{B}\hat{\alpha}-B\alpha^*\|_2^2\leq(\sin\theta(B,\hat{B}))^2\|B\alpha^*\|_2^2$.*

*Proof.*

$$\hat{B}\hat{\alpha}-B\alpha^*$$
$$=\hat{B}(\mathbf{E}[\hat{B}^\top xx^\top \hat{B}]^{-1})\hat{B}^\top \mathbf{E}[xx^\top B\alpha^*]-B\alpha^*$$
$$=\hat{B}(\mathbf{E}[\hat{B}^\top xx^\top \hat{B}]^{-1})\hat{B}^\top \mathbf{E}[xx^\top]B\alpha^*-B\alpha^*$$
$$=\hat{B}(\mathbf{E}[\hat{B}^\top xx^\top \hat{B}]^{-1})\hat{B}^\top \mathbf{E}[xx^\top](\hat{B}\hat{B}^\top B\alpha^*+(I-\hat{B}\hat{B}^\top)B\alpha^*)-B\alpha^*$$
$$=\hat{B}(\mathbf{E}[\hat{B}^\top xx^\top \hat{B}]^{-1})\hat{B}^\top \mathbf{E}[xx^\top]\hat{B}\hat{B}^\top B\alpha^*-B\alpha^*$$
$$=\hat{B}\hat{B}^\top B\alpha^*-B\alpha^*$$
$$=\hat{B}\hat{B}^\top B\alpha^*-BB^\top B\alpha^*$$
$$=(\hat{B}\hat{B}^\top-BB^\top)B\alpha^*$$

Then

$$\|\hat{B}\hat{\alpha}-B\alpha^*\|_2^2$$
$$=(\hat{B}\hat{B}^\top-BB^\top)B\alpha^*$$
$$\leq(\sin\theta(B,\hat{B}))^2\|B\alpha^*\|_2^2$$

$\square$

Then we have $\mathbf{E}\left[\left(\langle\hat{\alpha},\hat{B}^\top x\rangle-y\right)^2\right]-\mathbf{E}\left[\left(\langle\alpha^*,B^\top x\rangle-y\right)^2\right]\leq h(n_1)^2\|B\alpha^*\|_2^2$.

Putting these together gives

$$\mathbf{E}\left[\left(\langle\hat{\alpha}^{priv},\hat{B}^\top x\rangle-y\right)^2\right]-\mathbf{E}\left[\left(\langle\alpha^*,B^\top x\rangle-y\right)^2\right]$$
$$\leq g(n_2)+h(n_1)^2\|B\alpha^*\|_2^2$$

For the generic result with $\gamma$ subspace error, we substitute $h(n_1) = \gamma$, or Theorem 5.2 to instantiate the bound with the method of moments estimator [28]. Substituting B.3 for $g(n_2)$ and taking a union bound over failure probabilities gives the result. $\qquad\square$

## D.2    Proof of Theorem 5.5

Here, we prove our result lower bounding the lowest possible transfer error achieved by any $(\varepsilon,\delta)$-DP algorithm in our single-task transfer setting. We denote the class of two-stage algorithms of interest as $\mathcal{M}_{2\mathrm{stg}}(\epsilon,\delta,\gamma)$. Each algorithm in our class satisfies the following:

1. The algorithm takes as input private dataset $\mathcal{S}$ of $n_2$ i.i.d. samples from task $\alpha_{t+1}$, along with a $\gamma$-accurate public estimate $\hat{B}$,
2. Projects the input data point $x \mapsto \hat{B}^\top x$ for any $(x,y)$ in the private dataset $\mathcal{S}$.
3. Algorithm outputs an estimate in the column space of $\hat{B}$.

Similarly, we can define a class of misspecified linear regression problem instances. We use $\mathcal{P}_{2\mathrm{stg}}(d,k,\gamma)$ to denote the following class of problem instances for the private task $t+1$:

1. The input product distribution over $(x,y)$ is given by the model in (2), and additionally the noise $\eta \sim \mathcal{N}(0,\sigma^2)$.
2. Let the true regression vector be $B\alpha_{t+1}$. A subspace $\hat{B}$ is known such that: $\sin\theta(\hat{B},B) \leq \gamma$. Also, $\alpha_{t+1} \in \mathbb{R}^k$.
3. The i.i.d. sampled dataset $\mathcal{S}$ from the above model satisifies: $\|x\|_2 \leq 1$ for every $x \in \mathcal{S}$.

In the above, both $B$ and $\hat{B}$ are $d \times k$ matrices with orthonormal columns, i.e., $B, \hat{B} \in \mathrm{Gr}_{k,d}(\mathbb{R})$, where, $\mathrm{Gr}_{k,d}(\mathbb{R})$ is the Grassmann manifold [66] and consists of the set of $k$-dimensional subspaces within an underlying $d$-dimensional space. Also, for both $\mathcal{M}_{2\mathrm{stg}}(\epsilon,\delta,\gamma), \mathcal{P}_{2\mathrm{stg}}(d,k,\gamma)$ we omit the dependence on $\hat{B}$, which is fixed. We note here that our proof works for any fixed $\hat{B}$.

Now, we are ready to restate our Theorem 5.5.

**Theorem 5.5** (Two-stage single-task private transfer lower bound). *Let $M$ be an $(\varepsilon,\delta)$-DP private algorithm where $\varepsilon \in (0,1)$, $\delta < 1/n^{1+\omega}$, $\omega > 0$, that takes as input: (i) publicly estimated subspace $\hat{B}_{\mathrm{pub}}$ from an oracle that only uses public samples; and (ii) a dataset $\mathcal{S}$ of $n_2$ private samples. For any such $M$, there exists a private problem instance given by the pair $(B,\alpha_{t+1})$ where $B \in \mathrm{Gr}_{k,d}(\mathbb{R}), \alpha_{t+1} \in \mathbb{R}^k$, $\sin\theta(B,\hat{B}_{\mathrm{pub}}) \leq \gamma$, and $\|B\alpha_{t+1}\|_2 \leq 1$, such that for $S$ sampled i.i.d. from this instance using the model in (2), we have:*

$$\mathbb{E}_M \mathbb{E}_{\mathcal{S}|B,\alpha_{t+1}} \mathbb{E}_{(x,y)|B,\alpha_{t+1}} (y - M(\mathcal{S},\hat{B}_{\mathrm{pub}})^\top x)^2 \qquad (9)$$

$$= \Omega\left(\left(\frac{k^2}{n_2^2\varepsilon^2} + \frac{k}{n_2}\right)(\sigma^2 + \gamma^2) + \gamma^2\right). \qquad (10)$$

*Proof.* Given the estimate $\hat{B}$, the goal is to lower bound the following minimax risk:

$$\inf_{M \in \mathcal{M}_{2\mathrm{stg}}(\epsilon,\delta,\gamma)} \sup_{B,\alpha_{t+1} \in \mathcal{P}_{2\mathrm{stg}}(d,k,\gamma)} \mathbb{E}_M \mathbb{E}_{\mathcal{S}|B,\alpha_{t+1}} \mathbb{E}_{(x,y)|B,\alpha_{t+1}} (y - M(\mathcal{S},\hat{B})^\top x)^2 \qquad (20)$$

Let us begin by defining the class of regression vectors constiuting the set of all possible $B\alpha_{t+1}$ that can be realized by a problem instance in $\mathcal{P}_{2\mathrm{stg}}(d,k,\gamma)$. Given some rank $k$ subspace defined by the matrix $\hat{B} \in \mathrm{Gr}_{k,d}(\mathbb{R})$, we define the following set of $d$-dimensional $\ell_2$ norm bounded vectors that are $\gamma \leq 1$ close to given $\hat{B}$:

$$\theta(B,\gamma) =: \left\{\theta \in \mathbb{R}^d : \theta = B\alpha_{t+1} \text{ for } (B,\alpha_{t+1}) \in \mathcal{P}_{2\mathrm{stg}}(d,k,\gamma)\right\} \qquad (21)$$

From the definition of the principal angles and $\mathcal{P}_{2\mathrm{stg}}(d,k,\gamma)$ it follows that for any $\theta \in \theta(B,\gamma)$:

$$\|\hat{B}\theta\|_2 \geq \sqrt{1-\gamma^2} \iff \|(I_d - \hat{B}\hat{B}^\top)\theta\|_2 \leq \gamma$$

We can break the above set $\Theta(\hat{B},\gamma)$ into disjoint sets: $\Theta(\hat{B},\gamma) = \coprod_{\rho \in [\sqrt{1-\gamma^2},1]} \Theta_\rho(\hat{B})$, where $\Theta_\rho(\hat{B})$ is defined as:

$$\Theta_\rho(\hat{B}) =: \left\{ \theta \in \Theta(\hat{B},\gamma) : \|\hat{B}\theta\|_2 = \rho \right\} \tag{22}$$

The above subclass of regression vectors results in a convenient subclass of problem instances class $\mathcal{P}_{2\text{stg}}(\rho)$. Just as we did for $\mathcal{P}_{2\text{stg}}(d,k,\gamma)$, we can define the following minimax risk for $\mathcal{P}_{2\text{stg}}(\rho)$.

$$\inf_{M \in \mathcal{M}_{2\text{stg}}(\epsilon,\delta,\gamma)} \sup_{B,\alpha_{t+1} \in \mathcal{P}_{2\text{stg}}(\rho)} \mathbb{E}_{\mathcal{S}|B,\alpha_{t+1}} \mathbb{E}_{(x,y)|B,\alpha_{t+1}} (y - M(\mathcal{S},\hat{B})^\top x)^2. \tag{23}$$

Based on the above definitions we get:

$$\inf_{M \in \mathcal{M}_{2\text{stg}}(\epsilon,\delta,\gamma)} \sup_{B,\alpha_{t+1} \in \mathcal{P}_{2\text{stg}}(d,k,\gamma)} \mathbb{E}_{\mathcal{S}|B,\alpha_{t+1}} \mathbb{E}_{(x,y)|B,\alpha_{t+1}} (y - M(\mathcal{S},\hat{B})^\top x)^2 \tag{24}$$

$$= \inf_{M \in \mathcal{M}_{2\text{stg}}(\epsilon,\delta,\gamma)} \sup_{\rho \in [\sqrt{1-\gamma^2},1]} \sup_{B,\alpha_{t+1} \in \mathcal{P}_{2\text{stg}}(\rho)} \mathbb{E}_{\mathcal{S}|B,\alpha_{t+1}} \mathbb{E}_{(x,y)|B,\alpha_{t+1}} (y - M(\mathcal{S})^\top x)^2 \tag{25}$$

$$= \inf_{M \in \mathcal{M}_{2\text{stg}}(\epsilon,\delta,\gamma)} \sup_{\rho \in [\sqrt{1-\gamma^2},1]} \sup_{\theta \in \Theta_\rho(\hat{B})} \mathbb{E}_{\mathcal{S}|\theta=B\alpha_{t+1}} \mathbb{E}_{(x,y)|\theta=B\alpha_{t+1}} (y - M(\mathcal{S})^\top x)^2 \tag{26}$$

$$\geq \sup_{\rho \in [\sqrt{1-\gamma^2},1]} \inf_{M \in \mathcal{M}_{2\text{stg}}(\epsilon,\delta,\gamma)} \sup_{\theta \in \Theta_\rho(\hat{B})} (y - M(S)^\top x)^2, \tag{27}$$

where the final inequality uses $\inf \sup \geq \sup \inf$ [67]. We can do this because $\inf$ and $\sup$ are defined over non-empty sets and the loss function remains bounded over the product space $\mathcal{M}_{2\text{stg}}(\epsilon,\delta,\gamma) \times \Theta_\rho(\hat{B})$. The loss function is bounded because the norm of the regression vector and the input covariates is bounded. Further, the linearly independent noise $\eta$ in $y$ (2) has finite variance.

For the next part of the proof, we focus on lower bounding the minimax risk in (23) when the adversary is searching over the set $\Theta_\rho(\hat{B})$. The lower bound for the minimax risk over this subclass is given by two parts: (i) statistical error rate that is suffered by any non-private algorithm for which we lower bound hypothesis testing lower bounds; and (ii) the risk suffered by any $(\varepsilon,\delta)$-DP private estimator which we lower bound by constructing a tracing attack. We will begin the proof for the latter part and then plug in standard statistical risk lower bounds.

The following Lemma D.2 (proven later) states a lower bound over the class $\mathcal{P}_{2\text{stg}}(\rho)$.

**Lemma D.2** (Lower bound for $\mathcal{P}_{2\text{stg}}(\rho)$)**.** *For any fixed $\hat{B}$ and any $(\varepsilon,\delta)$-DP private algorithm $M$ (where $0 < \varepsilon < 1$, $\delta < 1/n^{1+\omega}$ for some $\omega > 0$) that belongs to class $\mathcal{M}_{2\text{stg}}(\epsilon,\delta,\gamma)$, there exists a problem instance for the transfer task in the class $\mathcal{P}_{2\text{stg}}(\rho)$ such that for the $B,\alpha_{t+1}$ given by the problem instance:*

$$\mathbb{E}_M \mathbb{E}_{\mathcal{S}|B,\alpha_{t+1}} \mathbb{E}_{(x,y)|B,\alpha_{t+1}} (y - M(\mathcal{S},\hat{B})^\top x)^2 = \Omega\left( \left( \frac{k^2}{n_2^2 \varepsilon^2} + \frac{k}{n_2} \right)(\sigma^2 + 1 - \rho^2) + 1 - \rho^2 \right). \tag{28}$$

We can now come back to (27) and compute the supremum over $\rho$ after plugging in the lower bound in Lemma D.2. Since $\rho \geq \sqrt{1-\gamma^2}$, plugging in this value for $\rho$ in Lemma D.2, and from the minimax risk lower bound on $\mathcal{P}_{2\text{stg}}(d,k,\gamma)$ in (27), we obtain the result in Theorem 5.5.

$\square$

### D.2.1 Proof of Lemma D.2

*Proof.* The proof for the subclass lower bound relies upon re-parameterizing the problem instance as a $k$-dimensional linear regression in the problem, but now in the column span of $\hat{B}$.

Let the worst case in instance in $\mathcal{P}_{2\text{stg}}(\rho)$ be $\theta = B\alpha_{t+1}$, where $\|\hat{B}^\top\theta\|_2 = \rho$. We shall derive a low-dimensional linear regression problem posed by the the unknown worst case instance $\theta$, and the projected inputs: $x \mapsto \hat{B}^\top x$. Recall that the joint data distribution for private samples is given by:

$$x \sim \mathcal{N}(0, I_d), \tag{29}$$

$$y \,|\, x \sim \mathcal{N}(x^\top\theta, \sigma^2) \tag{30}$$

Additionally, we also recall that the learning algorithm is given $n_2$ private i.i.d. samples from the above distribution $\mathcal{S} =: \{(x_i, y_i)\}_{i=1}^n$. In addition, it is also given a rank $k$ matrix with orthonormal columns: $\hat{B} \in \mathrm{Gr}_{k,d}(\mathbb{R})$ that is close to the unknown low rank subspace $B$, i.e., $\sin\theta(\hat{B}, B) \leq \gamma \implies \|(I_d - \hat{B}\hat{B}^\top)B\|_2 \leq \gamma$. Next, we write each sample in $\mathcal{S}$ in terms of the projection of the regression vector $B\alpha$ on to the column span of $\hat{B}$, when the input $x$ is also projected in the following way: $x \mapsto \hat{B}x$:

$$
\begin{aligned}
z &\sim \mathcal{N}(0, \sigma^2) \\
y &= x^\top B\alpha + z; \\
&= x^\top((\hat{B}\hat{B}^\top)B\alpha + (I_d - \hat{B}\hat{B}^\top)B\alpha) + z; \\
&= x^\top(\hat{B}\hat{B}^\top B\alpha) + x^\top(I_d - \hat{B}\hat{B}^\top)B\alpha + z; \\
&= (x^\top\hat{B})\hat{\alpha} + x^\top(I_d - \hat{B}\hat{B}^\top)B\alpha + z,
\end{aligned}
\tag{31}
$$

where $\hat{\alpha} := \hat{B}^\top B\alpha$ is a $k-$dimensional vector in the column span of the given subspace $\hat{B}$.

For any output $M(\mathcal{S}, \hat{B})$ for an algorithm in $\mathcal{M}_{2\mathrm{stg}}(\epsilon, \delta, \gamma)$, from the independence of the two projections: $\hat{B}\hat{B}^\top x$ and $(I_d - \hat{B}\hat{B}^\top)x$ argued in Lemma C.1.:

$$
\begin{aligned}
&\mathbb{E}_{\mathcal{S}|B,\alpha}\mathbb{E}_{(x,y)|B,\alpha_{t+1}}(y - M(\mathcal{S}, \hat{B})^\top x)^2 \\
&= \mathbb{E}_{\mathcal{S}|B,\alpha}\mathbb{E}_x\mathbb{E}_\eta(x^\top\hat{B}\hat{B}^\top\theta + x^\top(I_d - \hat{B}\hat{B}^\top)\theta + \eta - M(\mathcal{S}, \hat{B})^\top x)^2 \\
&= \sigma^2 + \|(I_d - \hat{B}\hat{B}^\top)\theta\|_2^2 + \mathbb{E}_{\mathcal{S}|B,\alpha}\mathbb{E}_x\mathbb{E}_\eta(x^\top\hat{B}\hat{B}^\top\theta - M(\mathcal{S}, \hat{B})^\top x)^2
\end{aligned}
\tag{32}
$$

Since, the norm of $\theta$ in the nullspace of $\hat{B}$ can be chosen without affecting the hardness of the above rejection problem, the worst case problem instance will maximize the additive error by picking any component along the null space (note that direction along null space does not impact the regression error) that has the maximum norm of $1 - \rho^2$ (recall that for any $\theta \in \mathcal{P}_{2\mathrm{stg}}(\rho)$, $\|\theta\|_2 \leq 1$). Now, from (31) it follows that the i.i.d. samples in $\mathcal{S}$ for the worst case instance are drawn from the following low-dimensional linear regression model:

$$
\begin{aligned}
x &\sim \mathcal{N}(0, I_d) \\
x^{\hat{B}} &=: x^\top\hat{B} \\
y \mid x^{\hat{B}} &\sim \mathcal{N}((x^{\hat{B}})^\top\hat{\alpha}, \hat{\sigma}^2), \quad \text{where,} \ \hat{\sigma}^2 = \sigma^2 + 1 - \rho^2
\end{aligned}
\tag{33}
$$

From the above model in (33) and equivalence in (32), we have the following equality for the minimax risk over $\mathcal{P}_{2\mathrm{stg}}(\rho)$.

$$
\begin{aligned}
&\inf_{M \in \mathcal{M}_{2\mathrm{stg}}(\epsilon,\delta,\gamma)} \ \sup_{B,\alpha \in \mathcal{P}_{2\mathrm{stg}}(\rho)} \mathbb{E}_{\mathcal{S}|B,\alpha}\mathbb{E}_{(x,y)|B,\alpha}(y - M(\mathcal{S}, \hat{B})^\top x)^2 \\
&= \inf_{M \in \mathcal{M}_{2\mathrm{stg}}(\epsilon,\delta,\gamma)} \ \sup_{\substack{\hat{\alpha} = \hat{B}^\top B\alpha, \\ B\alpha \in \mathcal{P}_{2\mathrm{stg}}(\rho)}} \mathbb{E}_{\mathcal{S}|\hat{\alpha}}\mathbb{E}_{x|\hat{\alpha}}(\hat{\alpha}^\top x^{\hat{B}} - M(\mathcal{S}, \hat{B})^\top x^{\hat{B}}) + \sigma^2 + 1 - \rho^2 \\
&= \inf_{M \in \mathcal{M}_{2\mathrm{stg}}(\epsilon,\delta,\gamma)} \ \sup_{\substack{\hat{\alpha} = \hat{B}^\top B\alpha, \\ B\alpha \in \mathcal{P}_{2\mathrm{stg}}(\rho)}} \mathbb{E}_{\mathcal{S}|\hat{\alpha}}\|\hat{\alpha} - M(\mathcal{S}, \hat{B})\|_2^2 + \sigma^2 + 1 - \rho^2,
\end{aligned}
\tag{34}
$$

where $\mathcal{S} = \{(x_i^{\hat{B}}, y_i)\}_{i=1}^{n_2}$ and $y_i = \hat{\alpha}^\top x_i^{\hat{B}} + z_i$, where $z_i \sim \mathcal{N}(0, \sigma^2 + 1 - \rho^2)$.

Our main technique for proving lower bounds for the minimax risk in (34) is based on the *tracing adversary* technique proposed by [59]. Next, we prove that there exists a prior over the effective regression vector $\hat{\alpha}$, and a tracing attack that is successful in recovering an element of the i.i.d. sampled dataset $\mathcal{S}$, in expectation over the prior and the dataset. Consider the following tracing attack:

$$
A_{\hat{\alpha}}((x^{\hat{B}}, y), M(\mathcal{S}, \hat{B})) = (y - (x^{\hat{B}})^\top\hat{\alpha}) \sum_{j=1}^{k-1}(M(\mathcal{S}, \hat{B})_j - \hat{\alpha}_j) \cdot x_j^{\hat{B}}.
\tag{35}
$$

Similar to the tracing attacks for mean estimation problems [58], Lemma D.3 proves that the attack $A_{\hat{\alpha}}((y, x^{\hat{B}}), M(\mathcal{S}))$ takes large value when $(x^{\hat{B}}, y)$ belongs to $\mathcal{S}$ and small value otherwise. We

compare the attack success with estimation error, and show that whenever the estimation error is small, the attack has to be fairly successful. Since, we are only searching over private algorithms where the attack takes small values, this yields a lower bound on the estimation error.

The minimax lower bound stated in Lemma D.2, i.e., the lower bound for the minimax risk over subclass $\mathcal{P}_{2\mathrm{stg}}(\rho)$ (for a fixed $\rho$), is given by the summation over two terms: the statistical lower bound and a second term implied by the tracing attack sucess lower bound stated in Lemma D.3 (proven later).

**Lemma D.3.** *For any fixed $0 < \sigma$, $\sqrt{1-\gamma^2} \leq \rho \leq 1$, $(B,\alpha)$ satisfying $\sin\ \theta(\hat{B},B) \leq \gamma$, and $\|\hat{\alpha}\|_2 = \|\hat{B}^\top B\alpha\|_2 = \rho \leq 1$, let $(x^{\hat{B}},y)$ be an i.i.d. sample (and $\mathcal{S}$ a dataset of $n_2$ i.i.d. samples) drawn from the distribution defined in (33). Then, for every $(\varepsilon,\delta)$-differentially private estimator $M$ that takes as input $\mathcal{S},\hat{B}$ and satisfies $\mathbb{E}_{\mathcal{S}|B,\alpha,\sigma,\rho}\|M(\mathcal{S},\hat{B})-\hat{\alpha}\|_2^2 = o(1)$, for every $\hat{\alpha}$, the following are true:*

1. *For each $i \in [n]$, let $\mathcal{S}_i'$ denote the data set obtained by replacing $(x_i^{\hat{B}},y_i)$ in $\mathcal{S}$ with an independent copy from the distribution in (31), then $\mathbb{E}A_{\hat{\alpha}}((x_i^{\hat{B}},y_i),M(\mathcal{S}_i'))=0$ and*

$$\mathbb{E}\,|A_{\hat{\alpha}}((x_i^{\hat{B}},y_i),M(\mathcal{S}_i',\hat{B}))| \leq (\sqrt{\sigma^2+1-\rho^2})\cdot\sqrt{\mathbb{E}\|M(\mathcal{S},\hat{B})-\hat{\alpha}\|_2^2}.$$

2. *There exists a prior distribution of $\pi=\pi(\hat{\alpha})$ supported over $\hat{\alpha}\in\mathbb{R}^k$ such that $\hat{\alpha}=\rho$, and*

$$\sum_{i\in[n]}\mathbb{E}_{\hat{\alpha}\sim\pi}\mathbb{E}_{\mathcal{S}|\hat{\alpha},\rho,\sigma}A_{\hat{\alpha}}((x_i^{\hat{B}},y_i),M(\mathcal{S}_i,\hat{B})) \gtrsim (\sigma^2+1-\rho^2)\cdot(k-1).$$

From Lemma C.2 and from the first part of Lemma D.3,

$$\sum_{i\in[n]}\mathbb{E}_{\mathcal{S}|\hat{\alpha}}A_{\hat{\alpha}}((x_i^{\hat{B}},y_i),M(\mathcal{S},\hat{B})) \leq 2n_2\varepsilon\sqrt{\sigma^2+1-\rho^2}\sqrt{\mathbb{E}_{\mathcal{S}|\hat{\alpha}}\|M(\mathcal{S},\hat{B})-\hat{\alpha}\|_2^2}$$

$$+2n_2\delta T+n_2\int_T^\infty \mathbb{P}\Big(|A_{\hat{\alpha}}((x_i^{\hat{B}},y_i),M(\mathcal{S},B))|>t\Big).$$

For the tail probability term,

$$\mathbb{P}\Big(|A_{\hat{\alpha}}((x_i^{\hat{B}},y_i),M(\mathcal{S},\hat{B}))|>t\Big)=\mathbb{P}\left(|y_i-x_i^\top\hat{\alpha}|\left|\sum_{j=1}^{k-1}(M(\mathcal{S},\hat{B})_j-\hat{\alpha}_j)\cdot x_j^{\hat{B}}\right|>t\right)$$

$$\leq\mathbb{P}\Big(|y_i-x_i^\top\hat{\alpha}|\|\hat{\alpha}\|\|x^{\hat{B}}\|>t\Big)$$

$$\leq\mathbb{P}\Big(|y_i-x_i^\top\hat{\alpha}|\sqrt{k}>t\Big)\leq 2\exp\left(\frac{-t^2}{2k(\sigma^2+1-\rho^2)}\right).$$

By choosing $T=\sqrt{2(\sigma^2+1-\rho^2)k\log(1/\delta)}$, we obtain

$$\sum_{i\in[n]}\mathbb{E}_{\mathcal{S}|\hat{\alpha}}A_{\hat{\alpha}}((x_i^{\hat{B}},y_i),M(\mathcal{S},\hat{B})) \lesssim 2n_2\varepsilon\sqrt{\sigma^2+1-\rho^2}\sqrt{\mathbb{E}_{\mathcal{S}|\hat{\alpha}}\|M(\mathcal{S},\hat{B})-\hat{\alpha}\|_2^2}$$

$$+\mathcal{O}\Big(n_2\delta\sqrt{(\sigma^2+1-\rho^2)k\log(1/\delta)}\Big).$$

Now plugging in the second part of Lemma D.3 gives us

$$(\sigma^2+1-\rho^2)k \leq \mathbb{E}_\pi\sum_{i\in[n]}\mathbb{E}_{\hat{\alpha}\sim\pi}\Big[\mathbb{E}_{\mathcal{S}|\hat{\alpha}}A_{\hat{\alpha}}((x_i^{\hat{B}},y_i),M(\mathcal{S},\hat{B}))\Big]$$

$$\lesssim 2n_2\varepsilon\sqrt{\sigma^2+1-\rho^2}\sqrt{\mathbb{E}_\pi\mathbb{E}_{\mathcal{S}|\hat{\alpha}}\|M(\mathcal{S})-\hat{\alpha}\|_2^2}+\mathcal{O}\Big(n_2\delta\sqrt{(\sigma^2+1-\rho^2)k\log(1/\delta)}\Big).$$

Since $\delta < n^{-(1+\omega)}$ for $\omega > 0$, for every $(\varepsilon,\delta)$-differentially private $M$ we have

$$\mathbb{E}_\pi\mathbb{E}_{\mathcal{S}|\hat{\alpha}}\|M(\mathcal{S})-\hat{\alpha}\|_2^2 \gtrsim (\sigma^2+1-\rho^2)\frac{k^2}{n_2^2\varepsilon^2}. \tag{36}$$

Adding the statistical lower bound of $\frac{k(\sigma^2+1-\rho^2)}{n_2}$ to the lower bound from (36), and from (34), we complete the proof of Lemma D.2.

$\square$

### D.2.2 Proof of Lemma D.3

*Proof.* Let us begin by looking at $\mathbb{E}A_{\hat{\alpha}}((x_i^{\hat{B}}, y_i), M(\mathcal{S}_i', \hat{B}))$, where we use the fact that $y_i - (x_i^{\hat{B}})^\top \hat{\alpha}$ is independent of $x_i^{\hat{B}}$, and $\mathbb{E}[y_i - (x_i^{\hat{B}})] = 0$:

$$\mathbb{E}A_{\hat{\alpha}}((x_i^{\hat{B}}, y_i), M(\mathcal{S}_i', \hat{B}))$$

$$= \mathbb{E}\left[ (y_i - (x_i^{\hat{B}})^\top \hat{\alpha}) \sum_{j=1}^{k-1} (M(\mathcal{S}_i', \hat{B})_j - \hat{\alpha}_j) x_{i,j}^{\hat{B}} \right]$$

$$= \mathbb{E}\left[ (y_i - (x_i^{\hat{B}})^\top \hat{\alpha}) \right] \sum_{j=1}^{k-1} \mathbb{E}[M(\mathcal{S}_i', \hat{B}) - \hat{\alpha}_j] \mathbb{E}[x_{i,j}^{\hat{B}}]$$

$$= 0 \cdot \sum_{j=1}^{k-1} \mathbb{E}[M(\mathcal{S}_i', \hat{B}) - \hat{\alpha}_j] \mathbb{E}[x_{i,j}^{\hat{B}}] = 0$$

This proves the first claim about the expected value of the attack when the datapoint is not a part of the training set, i.e., $\mathbb{E}A_{\hat{\alpha}}((x_i^{\hat{B}}, y_i), M(\mathcal{S}_i')) = 0$. Next, we look at the expected magnitude of the same random variable and upper bound it with a term that scales with the estimation error.

$$\mathbb{E}|A_{\hat{\alpha}}((x_i^{\hat{B}}, y_i), M(\mathcal{S}_i', \hat{B}))|$$

$$\leq \sqrt{\mathbb{E}(A_{\hat{\alpha}}((x_i^{\hat{B}}, y_i), M(\mathcal{S}_i')))^2} \quad \text{(Jensen's inequality)}$$

$$\leq \sqrt{\mathbb{E}\left[ \left( (y - (x^{\hat{B}})^\top \hat{\alpha}) \sum_{j=1}^{k-1} (M(\mathcal{S}, \hat{B})_j - \hat{\alpha}_j) \cdot x_j^{\hat{B}} \right)^2 \right]}$$

$$\leq \sqrt{\mathbb{E}\left[ \left( \langle M(\mathcal{S}_i', \hat{B}) - \hat{\alpha}, (y_i - (x_i^{\hat{B}})^\top \hat{\alpha}) x_i^{\hat{B}} \rangle^2 \right) \right]}$$

$$= \sqrt{\mathbb{E}[((y_i - (x_i^{\hat{B}})^\top \hat{\alpha}))^2 \cdot (M(\mathcal{S}_i', \hat{B}) - \hat{\alpha})^\top \mathbb{E}[(x_i^{\hat{B}})((x_i^{\hat{B}}))^\top](M(\mathcal{S}_i', \hat{B}) - \hat{\alpha})]} \quad \text{(independence)}$$

$$= \sqrt{\mathbb{E}\left[ ((y_i - (x_i^{\hat{B}})^\top \hat{\alpha}))^2 \cdot (M(\mathcal{S}_i', \hat{B}) - \hat{\alpha})^\top I_k (M(\mathcal{S}_i', \hat{B}) - \hat{\alpha}) \right]} \quad \text{(since } \hat{B}^\top \hat{B} = I_k\text{)}$$

$$= \sqrt{\sigma^2 + (1 - \rho^2)} \cdot \sqrt{\mathbb{E}\|M(\mathcal{S}_i', \hat{B}) - \hat{\alpha}\|_2^2}, \quad \text{(independence)}$$

where the last inequality uses the following derivation:

$$\mathbb{E}\left[ ((y_i - (x_i^{\hat{B}})^\top \hat{\alpha}))^2 \right] = \hat{\sigma}^2 = \sigma^2 + \|(I_d - \hat{B}\hat{B}^\top)B\alpha\|_2^2$$

$$= \sigma^2 + 1 - \rho^2$$

This completes the proof for the first part of the Lemma. For the second part we will begin by constructing a convenient prior for $\hat{\alpha}$.

Note that $\hat{\alpha}$ can take any value in the column span of $\hat{B}$ if the adversary has complete control over $B$ and $\alpha$. Thus, defining a prior over $\hat{\alpha}$ would involve defining a prior over the column span of $\hat{B}$ such that $\|\hat{\alpha}\|_2 = \rho$. We define a sample from the prior $\pi$ as a multi step procedure:

1. For all $i \in [k-1]$, sample $\omega_i$ from the truncated Gaussian, with mean 0, variance $\rho^2/(k-1)$, and truncation at points $-\rho/\sqrt{k-1}$ and $\rho/\sqrt{k-1}$.

2. Set $\omega_k = \pm\sqrt{1 - \sum_{i \in [k-1]} \omega_i^2}$ with equal probability for either sign.

3. Now, set $B\alpha = \sum_{i \in [k]} \omega_i \cdot v_i$, where $v_i$ is the $i^{\text{th}}$ column of $\hat{B}$. Consequently, $\hat{\alpha} = \hat{B}^\top B\alpha = [\omega_1, \omega_2, ..., \omega_k]^\top$.

For the second part of the claim we need to lower bound $\sum_{i \in [n]} \mathbb{E}_{\hat{\alpha} \sim \pi} \mathbb{E}\left[ A_{\hat{\alpha}}((y_i, x_i^{\hat{B}}), M(\mathcal{S})) | \hat{\alpha} \right]$ which we can decompose over co-ordinates in the following way:

$$
\sum_{i \in [n]} \mathbb{E}_{\hat{\alpha} \sim \pi} \mathbb{E}\left[ A_{\hat{\alpha}}((y_i, x_i^{\hat{B}}), M(\mathcal{S}, \hat{B})) | \hat{\alpha} \right]
$$

$$
= \sum_{j \in [k-1]} \mathbb{E}_{\hat{\alpha} \sim \pi} \mathbb{E}\left[ M(\mathcal{S}, \hat{B})_j \left( \sum_{i \in [n]} (y_i - \hat{\alpha}^\top x_i^{\hat{B}}) x_{i,j}^{\hat{B}} \right) | \hat{\alpha} \right]
$$

$$
= \sum_{j \in [k-1]} \mathbb{E}_{\hat{\alpha} \sim \pi} \mathbb{E}\left[ M(\mathcal{S}, \hat{B})_j \frac{\partial}{\partial \hat{\alpha}_j} [\log p(\mathcal{S} | B, \hat{\alpha}, \sigma)] (\sigma^2 + 1 - \rho^2) | \hat{\alpha} \right]
$$

$$
= (\sigma^2 + 1 - \rho^2) \cdot \sum_{j \in [k-1]} \mathbb{E}_{\hat{\alpha} \sim \pi} \mathbb{E}\left[ M(\mathcal{S}, \hat{B})_j \frac{\partial}{\partial \hat{\alpha}_j} [\log p(\mathcal{S} | B, \hat{\alpha}, \sigma)] | \hat{\alpha} \right]
$$

$$
= (\sigma^2 + 1 - \rho^2) \cdot \sum_{j \in [k-1]} \mathbb{E}_{\hat{\alpha} \sim \pi} \frac{\partial}{\partial \hat{\alpha}_j} \mathbb{E}\left[ M(\mathcal{S}, \hat{B})_j \right], \tag{37}
$$

where the final equation uses the log-derivative trick.

Next, we focus on $\mathbb{E}_{\hat{\alpha} \sim \pi} \left[ \frac{\partial}{\partial \hat{\alpha}_j} \mathbb{E}[M(\mathcal{S}, \hat{B})_j] \right]$ for any $j \in [k-1]$. Recall, that for any dimension $j \in [k-1]$, the prior $\pi$ draws a sample from the Gaussian $\mathcal{N}(0, \rho^2/{k-1})$, truncated at $-\rho/\sqrt{k-1}, \rho/\sqrt{k-1}$ independently. We will now apply Stein's Lemma (see Lemma C.3) for the term $\mathbb{E}_{\hat{\alpha} \sim \pi} \left[ \frac{\partial}{\partial \hat{\alpha}_j} \mathbb{E}[M(\mathcal{S}, \hat{B})_j] \right]$.

Denoting $\hat{\alpha}_{-j}$ as the set $\{\hat{\alpha}_j\}_{j=1}^k \setminus \{\hat{\alpha}_j\}$, and $\pi_j$ as the marginal prior over $j^{\text{th}}$ dimension of $\hat{\alpha}_j$, we can lower bound $\mathbb{E}_{\hat{\alpha} \sim \pi} \left[ \frac{\partial}{\partial \hat{\alpha}_j} \mathbb{E}[M(\mathcal{S})_j] \right]$ in the following way:

$$
\mathbb{E}_{\hat{\alpha} \sim \pi} \left[ \frac{\partial}{\partial \hat{\alpha}_j} \mathbb{E}[M(\mathcal{S}, \hat{B})_j] \right] = \mathbb{E}_{\hat{\alpha}_{-j}} \left[ \mathbb{E}_{\hat{\alpha}_j} \frac{\partial}{\partial \hat{\alpha}_j} \mathbb{E}_{\mathcal{S}}[M(\mathcal{S}, \hat{B})_j] | \hat{\alpha}_{-j} \right]
$$

$$
= \mathbb{E}_{\hat{\alpha}} \left[ -\frac{\pi_j'(\hat{\alpha}_j)}{\pi_j(\hat{\alpha}_j)} \mathbb{E}_{\mathcal{S}}[M(\mathcal{S}, \hat{B})_j] \right]
$$

$$
= \mathbb{E}_{\hat{\alpha}} \left[ -\frac{\pi_j'(\hat{\alpha}_j)}{\pi_j(\hat{\alpha}_j)} \mathbb{E}_{\mathcal{S}}[M(\mathcal{S}, \hat{B})_j - \hat{\alpha}_j + \hat{\alpha}_j] \right]
$$

$$
\geq \mathbb{E}_{\hat{\alpha}} \left[ -\hat{\alpha}_j \frac{\pi_j'(\hat{\alpha}_j)}{\pi_j(\hat{\alpha}_j)} \right] - \mathbb{E}_{\hat{\alpha}} \left[ |\pi_j'(\hat{\alpha}_j)/\pi_j(\hat{\alpha}_j)| \cdot \mathbb{E}_{\mathcal{S}}[|M(\mathcal{S}, \hat{B})_j - \hat{\alpha}_j|] \right] \tag{38}
$$

Next, we use the density of the truncated Normal:

$$
\pi_j(\hat{\alpha}_j) = \frac{\exp\left(-\frac{(k-1)}{2\rho^2} \cdot \hat{\alpha}_j^2\right)}{\sqrt{2\pi}\rho/\sqrt{k-1} \cdot (\Phi(1) - \Phi(-1))},
$$

where $\Phi(\cdot)$ is the CDF function for a standard Normal distribution. Thus, $\frac{\pi_j'(\hat{\alpha}_j)}{\pi_j(\hat{\alpha}_j)} = -\frac{(k-1)}{\rho^2} \hat{\alpha}_j$.

Substituting the above and applying Cauchy-Schwarz followed by Jensen's inequality we get,

$$\sum_{j=1}^{k-1} \mathbb{E}_{\hat{\alpha}}\Big[|\pi_j'(\hat{\alpha}_j)/\pi_j(\hat{\alpha}_j)|\cdot\mathbb{E}_{\mathcal{S}}[|M(\mathcal{S},\hat{B})_j-\hat{\alpha}_j|]\Big]$$

$$= {}^{(k-1)}/\rho^2\cdot\mathbb{E}_{\hat{\alpha}}\left[\sum_{j=1}^{k-1}|\hat{\alpha}_j|\cdot\mathbb{E}_{\mathcal{S}}[|M(\mathcal{S},\hat{B})_j-\hat{\alpha}_j|]\right]$$

$$\le {}^{(k-1)}/\rho^2\cdot\sqrt{\mathbb{E}_{\hat{\alpha}}\left[\sum_{j\in[k-1]}\hat{\alpha}_j^2\right]\cdot\mathbb{E}_{\hat{\alpha}}\left[\sum_{j\in[k-1]}\Big(\mathbb{E}_{\mathcal{S}}\Big[M(\mathcal{S},\hat{B})_j-\hat{\alpha}_j\Big]\Big)^2\right]}$$

$$\le {}^{(k-1)}/\rho^2\cdot\sqrt{\mathbb{E}_{\hat{\alpha}}\|\hat{\alpha}\|^2}\sqrt{\mathbb{E}_{\hat{\alpha}}\mathbb{E}_{\mathcal{S}}\|M(\mathcal{S},\hat{B})_j-\hat{\alpha}\|^2} \tag{39}$$

From directly applying the density of the truncated Normal distribution we get,

$$\sum_{j=1}^{k-1}\mathbb{E}_{\hat{\alpha}}\left[-\hat{\alpha}_j\frac{\pi_j'(\hat{\alpha}_j)}{\pi_j(\hat{\alpha}_j)}\right] = {}^{(k-1)}/\rho^2\cdot\mathbb{E}_{\hat{\alpha}}\sum_{j\in[k-1]}\hat{\alpha}_j^2 \tag{40}$$

Plugging (40), (39) into (38), and using (37) we get,

$$\sum_{i\in[n]}\mathbb{E}_{\hat{\alpha}\sim\pi}\mathbb{E}\Big[A_{\hat{\alpha}}((y_i,x_i^{\hat{B}}),M(\mathcal{S}))|\hat{\alpha}\Big]$$

$$\ge \frac{(\sigma^2+1-\rho^2)}{\rho^2/(k-1)}\cdot\left(\mathbb{E}_{\hat{\alpha}\sim\pi}\sum_{j=1}^{k-1}\hat{\alpha}_j^2-\sqrt{\mathbb{E}_{\hat{\alpha}\sim\pi}\mathbb{E}_{\mathcal{S}|\hat{\alpha}}\|M(\mathcal{S},\hat{B})-\hat{\alpha}\|_2^2}\sqrt{\mathbb{E}_{\hat{\alpha}\sim\pi}\|\hat{\alpha}\|_2^2}\right) \tag{41}$$

Note that $\mathbb{E}_{\hat{\alpha}\sim\pi}\sum_{j=1}^{k-1}\hat{\alpha}_j^2=\rho^2$ by construction of the prior $\pi$ and $\mathbb{E}_{\hat{\alpha}\sim\pi}\mathbb{E}_{\mathcal{S}|\hat{\alpha}}\|M(\mathcal{S},\hat{B})-\hat{\alpha}\|_2^2=o(1)$ by assumption. Thus, $\sum_{i\in[n]}\mathbb{E}_{\pi}\mathbb{E}_{\mathcal{S}|B,\hat{B},\alpha,\sigma}A_{\hat{\alpha}}((x_i^{\hat{B}},y_i),M(\mathcal{S}_i,\hat{B})) \gtrsim (\sigma^2+1-\rho^2)\cdot(k-1)$, which completes the proof of the second claim in Lemma D.3.

$\square$

