# OpenReview forum: "On the Benefits of Public Representations for Private Transfer Learning under Distribution Shift"
_NeurIPS.cc/2024/Conference — NeurIPS 2024 poster_

### Official Review · Reviewer_iSuQ · 2024-07-11

**Soundness:** 3
**Presentation:** 3
**Contribution:** 4
**Rating:** 7
**Confidence:** 3

**Summary:**

The work aims to bridge an important gap in literature, looking at using public pretraining to improve differentially private model training, even when there is a distribution shift, particularly concept shift, in the public and private data. They report impressive improvements in accuracy. Further, they provide a justification for these observations, citing the representation sharing nature of the private and public data.

**Strengths:**

- [S1] Highlighting the limitations of current public pre-training techniques which have the same distribution as the private data, which as the authors rightly point out, is often not the case. There is a reason why the data is private and one would suspect, more often than not, there would be differences in the nature of the public and private data.

- [S2] Looks at the most general type of distribution shift, namely concept shift over covariate shift or label shift.

- [S3] Highlight on cases where models when trained from scratch privately collapse, and where there is an actual need for additional ways to supply information. Line 155-156 clearly distinguishes between situations where zero-shot private performance translates quite well, and thus the task itself is of lesser interest to study from a research perspective as the headroom is not as significant.

- [S4] I personally like the writing style of the paper quite a bit, it establishes clearly the gaps in the current research and even the setups people use, and goes on to very thoroughly present an alternative direction that should attract more work.

- [S5] I further like the presentation of theorems that is supplemented with Proof Sketches that help grasp the intuition of the proof.

**Weaknesses:**

- [W1] Strong assumption that the shared, low-dimensional representation is even learnable across the private and public data.

**Questions:**

- [Q1] Referencing W1 it might be interesting to look at the naturally learnt representations, when the public data and private data is used independently to train different models. It would help verify the assumption that a common representation space is even learnable. What kind of representations are learnt? What is the difference between the representations learnt in a disjoint way and those learnt by forcing a common latent space?

**Limitations:**

All the limitations, if any, have been adequately discussed and addressed.

---

> ### Author Rebuttal · Authors · 2024-08-07
>
> We thank the reviewer for their feedback and for the kind comments regarding the strengths of the work.
>
> > **[W1] Strong assumption that the shared, low-dimensional representation is even learnable across the private and public data.**
>
> > **[Q1] Referencing W1 it might be interesting to look at the naturally learnt representations, when the public data and private data is used independently to train different models. It would help verify the assumption that a common representation space is even learnable. What kind of representations are learnt? What is the difference between the representations learnt in a disjoint way and those learnt by forcing a common latent space?**
>
> The reviewer raises an interesting point regarding how realistic the shared low-rank subspace assumption is in practice, for deep models and real data. Answering these questions in the most rigorous way may be an interesting paper in itself; this would require defining a metric to measure similarity between trained models (or containment in a subspace) to distinguish the pretrained representation from the trained-from-scratch representation. Another subtlety is that the private trained-from-scratch representations in our setting are simply unusable, and are likely not representative of the "true" representation of the data—this raises a question of whether we should compare to a private or non-privately trained representation in order to measure containment in a subspace.
>
> That said, as a preliminary exploration, we plotted the eigenspectrum of the feature covariance matrix computed after extracting features of PCam images from the ViT-B-32 pretrained model. *This result is attached in the PDF in the global rebuttal.*
>
> From these results, we see that the pretrained features are approximately low-rank for the out-of-distribution task PCam, yet a linear probe over these features achieves good (83.5%) performance (Table 1). The fact that the representation still gives good performance when only a linear layer is trained on top suggests that the data does fundamentally lie in or near the low-rank space that is identified by the pretrained model. We would be happy to also provide this analysis for RESISC45 and fMoW in our revision.

---

> ### Comment · Reviewer_iSuQ · 2024-08-08
> **Response to Rebuttal**
>
> - I appreciate the authors' effort to undertake the plotting exercise in the short rebuttal deadline. The outcome on PCam does look reassuring! The inclusion of the plots for eigenspectrum for RESISC45 and fMoW would help round off the paper quite well.
>
> - I further agree that the measurement of distance between two representations is not quite obvious, especially for the high dimensional ones. A very very recent ICML paper (that I don't expect the authors to discuss at all now, but could be interesting for the future) looks into this problem it seems, and does a reasonably good job of providing measures to compare them. [1]
>   - Also, as the privately trained representations are in fact not usable is a valid observation. I would personally still err on the side of comparison against the private one to understand what differences arise when using public pre-training, and how the change in representation helps the model learn better.
>   - On the other hand, comparing representations against the well performing non-private training ones, could act like the other side of the spectrum?
>   - I'm curious to see which side the public-pretraining but privately trained models lie in this hyper-spectrum (if they do even lie somewhere useful), or do they lie in a completely unrelated subspace to either of these in the latent space. Something as simple as representing the latent representations as vectors for some smaller (maybe even toy) datasets and then analyzing them could be a good starting point.
>   - But then again, I do agree that this might require rigorous definitions in themselves and is too involved to be discussed in this paper. But I do find this direction fascinating and hope the authors' can follow up in a future version.
>
> - All in all, I've read the comments and feedback provided by the other reviewers', and am happy to recommend acceptance for the submission and keep my score :)
>
> [1] Wayland, Jeremy, Corinna Coupette, and Bastian Rieck. "Mapping the multiverse of latent representations." arXiv preprint arXiv:2402.01514 (2024).

---

### Official Review · Reviewer_PptC · 2024-07-14

**Soundness:** 3
**Presentation:** 3
**Contribution:** 3
**Rating:** 7
**Confidence:** 3

**Summary:**

There are many contexts where deep learning models trained to preserve differential privacy have much worse performance than models trained without the differential privacy constraint. One common way to improve model quality in these cases is to pre-train a model on publicly available data then fine-tune with the private data. These pre-trained models often demonstrate much better empirical performance than models trained exclusively on private data, even when the public data used for pre-training is very different from the private data. This phenomenon is not well understood theoretically so it is unclear what to expect from out-of-distribution pre-training.

This work seeks to understand public pre-training under distribution shift. The authors start by demonstrating the utility of public pre-training in the case of extreme distribution shift. Having demonstrated the phenomenon, the authors develop a theoretical model to explain the effectiveness of public pre-training. This model poses public data as coming from a sequence of linear regression tasks where the features of each task lie in a low-dimensional subspace. Working in this model, the authors develop a two stage algorithm for public-private linear regression that first approximates the subspace $B$ then uses DP-SGD to estimate the regression parameters $\alpha$. The authors analyze this algorithm to prove an upper bound on the error and they provide a matching lower bound among algorithms that solve the regression problem on a subspace. To support the theoretical results, the authors provide empirical results on simulated data.

**Strengths:**

- Analyzing the impact of public data can be difficult without making strong assumptions about the public data distribution, but it is frequently observed that pre-training/transfer learning with out of distribution data is beneficial. Bringing this distribution shift model over from the meta-learning literature seems like a promising approach for understanding this phenomenon.
-  The experiments on the vision datasets do a good job of demonstrating an clear example of OOD transfer learning being effective.
- The theoretical model is introduced and motivated well, the results are clearly stated and, while full proofs are in the appendix, the authors describe the proof strategy in the main text to give helpful intuition to the readers.
- The simulated result are good to see because they highlight the two sources of error (subspace estimation and DP-SGD error)

**Weaknesses:**

- The disconnect between the setting for the experiments and the stylized theoretical model is very apparent. It is difficult to say how much of the empirical results from section 4 can be explained by the theoretical results in section 5. That being said, the authors provide citations showing that this theoretical model is common in the meta-learning literature. I am not familiar enough with that literature to know if there are other alternate models that are common in that literature or how they would compare to the model used in this work. It would improve the paper if the authors compared to other potential models, if they exist, or stated that they do not.

**Questions:**

- The authors discuss public data assisted query answering in the background section, are you aware of [this](https://proceedings.mlr.press/v238/fuentes24a.html) more recent work in the area?
- An alternative to linear probing for parameter efficient fine tuning is [LoRA](https://arxiv.org/pdf/2106.09685), would you expect this method to perform similarly to linear probing in your transfer learning experiments?
- How does the bound for your two stage algorithm compare to the error you would attain if you simply ignored the public data? It would be useful to say exactly what you gain from the subspace estimation step and if the naive approach would ever be better (like in a situation where the embedding dimension is almost as large as the latent dimension).

**Limitations:**

- The authors have adequately addressed the main limitation of the work (the fact that the theoretical model is limited and does not totally account for the neural representations used in practice). They accurately describe their model as being stylized and acknowledge the impacts of this limitation in the discussion section.

---

> ### Author Rebuttal · Authors · 2024-08-07
>
> We thank the reviewer for their thoughtful feedback and comments on our work.
>
> > **The disconnect between the setting for the experiments and the stylized theoretical model is very apparent. It is difficult to say how much of the empirical results from section 4 can be explained by the theoretical results in section 5. [...] It would improve the paper if the authors compared to other potential models, if they exist, or stated that they do not.**
>
> Indeed, some other models and theoretical analysis techniques do exist in the non-private meta-learning literature. For example, [d] analyzes the gradient dynamics of model-agnostic meta-learning (MAML), and [e] proposes and analyzes a hierarchical clustering approach that uses multiple different MAML initializations to adapt well to new tasks.
>
> Nevertheless, as evidenced by the works cited in our paper (lines 97-101, 226-227), due to the tractable and simple nature of the linear subspace model, it has attracted more attention than other analyses even in the nonprivate meta-learning literature. As we mention in the common response, this model is particularly well-suited for understanding transfer learning under concept shift. We will include a more thorough discussion of other theoretical models from the non-private setting in our revision.
>
> > **The authors discuss public data assisted query answering in the background section, are you aware of this more recent work in the area?**
>
> Thank you for this reference; we were not aware of this work and will add it to our related work section.
>
> > **An alternative to linear probing for parameter efficient fine tuning is LoRA, would you expect this method to perform similarly to linear probing in your transfer learning experiments?**
>
> This is an interesting direction for future exploration. In our results, linear probing already achieves performance much higher than standard fine-tuning. LoRA would still involve training more parameters than linear probing, though fewer than full fine-tuning. In our experiments, we observe that full private fine-tuning performs worse than private linear probing, and one explanation for this is the larger number of parameters involved in full fine-tuning. Thus we would expect LoRA to be another point on the tradeoff between the degradation from increased dimension vs. the improvement from accuracy.
>
> > **How does the bound for your two stage algorithm compare to the error you would attain if you simply ignored the public data? It would be useful to say exactly what you gain from the subspace estimation step and if the naive approach would ever be better (like in a situation where the embedding dimension is almost as large as the latent dimension).**
>
> We point to the bound of Tripuraneni et al. (stated as Theorem 5.2 in our paper) as a starting point for this discussion. The error $\gamma$ of the subspace estimation algorithm given in that paper grows as $O(\sqrt{dk^2/n_1})$. As the reviewer suggested, this indicates that when the latent dimension is nearly as large as the embedding dimension, the subspace estimation error will outweigh the benefits of the dimension reduction. We would be happy to add a discussion about this in an updated version.

---

> > ### Comment · Reviewer_PptC · 2024-08-12
> >
> > Thank you to the authors for clearly answering my questions about the work. I have reviewed the rebuttal along with the comments from other reviewers and decided to stick with my original score.

---

### Official Review · Reviewer_t1VG · 2024-07-15

**Soundness:** 3
**Presentation:** 3
**Contribution:** 3
**Rating:** 6
**Confidence:** 3

**Summary:**

The paper proposes a theory for the benefit of transfer learning in DP ML even when public data and private data differ significantly from each other. This theory relies on a subspace shared between public and private data. They empirically verify the algorithm in their theory.

**Strengths:**

The paper is very clearly written and structured. It demonstrates a phenomenon, develops a theory to explain it, and verifies their theoretical results experimentally.

The paper makes a connection to theoretical work from metalearning.

The lower bound in Theorem 5.5 is interesting.

**Weaknesses:**

The advantage over prior work is mild for linear regression tasks. In linear regression, gradients, input data, or models being constrained to a linear subspace are all roughly the same thing.

The paper does not attempt to reconnect the theory back to the original experiments. This explanation does not seem refutable.

The paper does not quantitatively argue that there is limited overlap between the datasets considered and the base pretraining data.

**Questions:**

Can you verify that there is limited overlap between the benchmark datasets evaluated and the DataComp dataset? I think even some sort of textual overlap is sufficient if not exhaustive.

**Limitations:**

There is a fair discussion of limitations.

---

> ### Author Rebuttal · Authors · 2024-08-07
>
> We thank the reviewer for taking the time to read and provide feedback on our work.
>
> > **The advantage over prior work is mild for linear regression tasks. In linear regression, gradients, input data, or models being constrained to a linear subspace are all roughly the same thing.**
>
> We acknowledge that these three settings (gradient, input, or model/task subspaces) lead to similar analyses in the linear regression setting. That said, we focus on the shared subspace of the linear models in public and private data distributions because it cleanly models concept shift and shared representation. Such a model also has not been explicitly stated or studied in prior work on public-to-private transfer.
>
> > **The paper does not attempt to reconnect the theory back to the original experiments. This explanation does not seem refutable.**
>
> We understand the reviewer's concern regarding the disconnect between theory and practice. As we mention in the common response, our theory provides one simple model in which public data can indeed help under significant distribution shift (and one which has been validated to be useful empirically, in the nonprivate setting). That said, there is significant current research interest in understanding whether this "shared representation" model is accurate in practice.
>
> > **The paper does not quantitatively argue that there is limited overlap between the datasets considered and the base pretraining data [...] Can you verify that there is limited overlap between the benchmark datasets evaluated and the DataComp dataset? I think even some sort of textual overlap is sufficient if not exhaustive.**
>
> In our paper, we point to the low zero-shot performance of the base CLIP models on each dataset as a natural proxy for the absence of sufficient training data in the CLIP dataset. (In contrast, for example, CLIP has high zero-shot performance on ImageNet, which is contained in the training data, as well as on ImageNet variants [c].)
>
> For RESISC45 and fMoW, which are remote-sensing datasets, we also refer the reviewer to [b], which states: "We obtain a remote sensing subset of [LAION-2B] by applying a binary classification model to determine whether an image in LAION-2B is a remote sensing image (see details in Appendix A.5). This subset, denoted as LAION-RS, contains 726K remote sensing image-text pairs—only **0.03% of all samples.** This shows that web crawling cannot efficiently collect remote sensing image-text pairs at scale."
>
> We would be happy to replicate this analysis for PCam in an updated version of the paper (for Datacomp rather than LAION as LAION data is no longer publicly available). Unfortunately, due to the size and download time of the Datacomp dataset we were not able to perform this analysis during the rebuttal period.

---

> ### Author Response · Authors · 2024-08-12
>
> Hello, we would like to thank the reviewer again for their thoughtful feedback. We hope our responses have provided clarity on the concerns regarding the theoretical model and the composition of the datasets. We would like to gently follow up in case there are any further questions so that we have time to respond appropriately before the discussion period ends. Thank you!

---

### Official Review · Reviewer_S2j2 · 2024-07-17

**Soundness:** 3
**Presentation:** 3
**Contribution:** 2
**Rating:** 6
**Confidence:** 3

**Summary:**

This paper studies the role of using public training data for fine-tuning private tasks. The paper begins by showing, empirically, on three datasets, that fine-tuning significantly outperforms training privately from scratch. The experiments are conducted on several image classification datasets. Then, the paper provides a stylized theoretical model to explain the findings, based on models of nonprivate transfer learning. This model focuses on linear regression with isotropic noise corruption. The results are complemented by simulations matching the linear regression setup.

**Strengths:**

- The paper provides both theoretical and empirical evidence to show that public training data are useful for private fine-tuning, even when the two sources come from different data distributions.

- In the theoretical model, both upper and lower bounds are provided to justify the optimality of the rates.

**Weaknesses:**

- The theoretical setup may be too simplistic. For instance, the resulting algorithm 1 is different from what was implemented in the experiments.

- The experiments only focus on image classification. Additional data modality could be utilized to broaden the interest of these results.

**Questions:**

See Weaknesses above.

**Limitations:**

The authors discussed the limitations of their work in Section 6.

---

> ### Author Rebuttal · Authors · 2024-08-07
>
> We thank the reviewer for their time and feedback on our work.
>
> > **The theoretical setup may be too simplistic. For instance, the resulting algorithm 1 is different from what was implemented in the experiments.**
>
> We understand the reviewers' concerns regarding the stylized theoretical model. We aimed to analyze the simplest model that would capture the main characteristics of public pretraining as well as the key property of concept shift. As we note in the common response, this model is commonly used in the non-private meta-learning literature to model similar experiments in the non-private setting. Nevertheless, we agree that building off our work to explore a more complex model capturing the performance of public-to-private transfer would be an interesting direction of future work.
>
> > **The experiments only focus on image classification. Additional data modality could be utilized to broaden the interest of these results.**
>
> Thanks for this suggestion. We agree that experiments in another modality, such as language, could be interesting future work. Works such as [f, g] have shown that private training from scratch is ineffective for language models, while privately finetuning a public model can reach state-of-the-art (private) performance on standard NLP benchmarks. However, these works have not strictly explored the distribution shift setting. We expect that the benefits of a shared representation would extend across different language domains (due to the shared low-level features of language domains), but verifying this experimentally would be valuable. We will include a discussion of this in our revision.

---

> ### Author Response · Authors · 2024-08-12
>
> Hello, we would like to thank the reviewer again for their thoughtful feedback. We hope our responses have provided clarity on the concerns regarding the theoretical model and the experimental evaluation. We would like to gently follow up in case there are any further questions so that we have time to respond appropriately before the discussion period ends. Thank you!

---

### Author Rebuttal · Authors · 2024-08-07

## Common response to all reviewers

We thank all of the reviewers for taking the time to read and give helpful feedback on our paper.

As noted by the reviewers, our work provides both theoretical and empirical evidence to show that public pretraining is useful for private finetuning under distribution shift, and we appreciate the reviewers' acknowledgement of the significance of this work in the context of practical private machine learning in the realistic setting of concept shift.

In addition to providing individual responses to each reviewer, we would like to address here a common concern across reviewers regarding the relevance of the stylized theoretical model and the potential disconnect between the theoretical model and experimental setup.

We first note that the model we present is sufficient to answer the main research question we set out to answer: **Can public data help private learning under significant distribution shift?** Our model illustrates a clean, simple setting in which public data does in fact help, even when the private task can differ significantly. In this sense, we see the simplicity of the model as an advantage in that it highlights what we believe is a key benefit of using public data: learning the shared representation -- and answers our research question in the affirmative.

However, as we have touched on in our paper (lines 97-101, 226-227) we also note that analysis even in the non-private meta-learning setting is largely restricted to this linear subspace model. Out of the many examples we cite in our paper, we point to [a] as one example of work in the non-private setting studying a phenomenon similar to the one we study where the model is also restricted to a similar, linear subspace assumption to make the theoretical analysis tractable. Nevertheless, this work also finds that the simplified model is a good predictor for real experiments on fine-tuning tasks with deep networks. This again gives us confidence that the model is valuable to study even while being simple.

Finally, we point out that we in fact observe the biggest empirical improvement in the linear probing setting. As such, the linear regression model is particularly relevant because we learn a lower-dimensional linear model on top of the pretrained features, although those features may or may not be accurately represented by a low-rank linear subspace.



## Citations referenced in rebuttal (common to all responses)

[a] Kumar, A., Raghunathan, A., Jones, R., Ma, T., & Liang, P. (2022). Fine-tuning can distort pretrained features and underperform out-of-distribution. arXiv preprint arXiv:2202.10054.

[b] Wang, Z., Prabha, R., Huang, T., Wu, J., & Rajagopal, R. (2024, March). Skyscript: A large and semantically diverse vision-language dataset for remote sensing. In Proceedings of the AAAI Conference on Artificial Intelligence (Vol. 38, No. 6, pp. 5805-5813).

[c] Wortsman, M., Ilharco, G., Kim, J. W., Li, M., Kornblith, S., Roelofs, R., ... & Schmidt, L. (2022). Robust fine-tuning of zero-shot models. In Proceedings of the IEEE/CVF conference on computer vision and pattern recognition (pp. 7959-7971).

[d] Nichol, A., Achiam, J., & Schulman, J. (2018). On first-order meta-learning algorithms. arXiv preprint arXiv:1803.02999.

[e] Yao, H., Wei, Y., Huang, J., & Li, Z. (2019, May). Hierarchically structured meta-learning. In International conference on machine learning (pp. 7045-7054). PMLR.

[f] Yu, D., Naik, S., Backurs, A., Gopi, S., Inan, H. A., Kamath, G., ... & Zhang, H. (2021). Differentially private fine-tuning of language models. arXiv preprint arXiv:2110.06500.

[g] Li, X., Tramer, F., Liang, P., & Hashimoto, T. (2021). Large language models can be strong differentially private learners. arXiv preprint arXiv:2110.05679.

---

### Decision · Program_Chairs · 2024-09-25

**Decision:**

Accept (poster)

**Comment:**

The reviewers were overall positive about the paper. However, there were a lingering concern regarding the disconnect between the theory and the experiments. We recommend the authors to discuss this aspect in more detail in the paper, in order, to ensure a better readability of the paper. We do acknowledge the authors address this point a fair bit in the rebuttal. It would be great if some of the discussion from the rebuttal can be ported to the paper.